# Laboratory Evaluation of ARMIE, a Portable SPS30-Based Low-Cost Sensor Node for PM_2.5_ Monitoring

**DOI:** 10.3390/s26010280

**Published:** 2026-01-02

**Authors:** Asbjørn Kloppenborg, Louise B. Frederickson, Rasmus Ø. Nielsen, Clive E. Sabel, Tue Skallgaard, Jakob Löndahl, Jose G. C. Laurent, Torben Sigsgaard

**Affiliations:** 1Department of Public Health, Aarhus University, 8000 Aarhus C, Denmarkts@ph.au.dk (T.S.); 2Department of Environmental Science, Aarhus University, 4000 Roskilde, Denmark; 3Research Unit for General Practice in Aarhus, 8000 Aarhus C, Denmark; 4School of Geography, Earth and Environmental Sciences, University of Plymouth, Plymouth PL4 8AA, UK; 5Division of Ergonomics and Aerosol Technology, Lund University, 22100 Lund, Sweden; 6Department of Environmental and Occupational Health and Justice, Rutgers School of Public Health, Piscataway, NJ 08854, USA

**Keywords:** low-cost sensors, SPS30, PM_2.5_ monitoring, sensor calibration, performance evaluation, exposure assessment, environmental epidemiology

## Abstract

Background: Low-cost particulate matter sensors have enabled new opportunities for exposure monitoring but require evaluation before application in epidemiological studies. This study assessed the performance of the SPS30 sensor integrated into the ARMIE portable monitoring sensor-node under controlled laboratory conditions. Methods: Sensors were co-located with two comparison instruments—the optical DustTrak photometer and the combined Scanning Mobility Particle Sizer (SMPS) and Aerodynamic Particle Sizer (APS)—across multiple aerosol sources, including candle burning, cooking, cigarette smoke, and clean air, under both regular and high-humidity conditions. Calibration performance was evaluated using leave-one-sensor-out and leave-one-source-out approaches. Results: The ARMIE node demonstrated strong agreement with the DustTrak (*r* = 0.93–0.98) and maintained linear response characteristics across emission types. Calibration reduced mean errors and narrowed the limits of agreement. Agreement with the SMPS + APS was moderate (*r* = 0.74–0.94) and characterized by systematic underestimation at higher concentrations. Conclusions: Overall, the ARMIE node achieved high correlation with the DustTrak, demonstrating that low-cost optical sensors can reliably capture temporal variability in particle concentrations relative to mid-cost photometers.

## 1. Introduction

Air pollution remains a leading environmental determinant of premature morbidity and mortality worldwide. Fine particulate matter (PM_2.5_; particles 2.5 μm in diameter and smaller) alone accounts for millions of deaths annually and is a major contributor to the global burden of disease [1]. Epidemiological studies consistently link short-term fluctuations in PM_2.5_ to acute events, such as myocardial infarction and arrhythmias, which can occur within hours of exposure peaks [2]. However, most exposure assessments rely on fixed-site monitors, which may underestimate individual risk due to spatial and temporal heterogeneity of urban air pollution [3].

Individual exposure often diverges from ambient concentrations across microenvironments and daily activities [4]. Commuting, cooking, and burning candles or using e-cigarettes can produce disproportionately high personal exposures despite low background levels [5,6]. Because acute cardiovascular responses can occur within hours of exposure peaks, temporal resolution may be as important as spatial coverage in capturing risk-relevant fluctuations. This ambient–personal exposure misclassification may bias risk estimates by diminishing the importance of short-term exposure peaks. Therefore, high-resolution personal monitoring is essential to capture the true pattern and history of air pollution exposures [7].

Low-cost optical particulate matter (PM) sensors provide a practical means for direct personal exposure measurement, enabling a shift from sparse fixed-site monitoring to high-density networks and personal monitoring [8,9]. While reference instruments are accurate, their cost and size limit their use in large-scale deployments [10]. In contrast, low-cost optical PM sensors (e.g., Sensirion SPS30) correlate well with reference devices under controlled conditions, although accuracy depends on aerosol characteristics and humidity [11,12,13]. Recent syntheses highlight a need for calibration and contextual validation that is widespread across the field. A comprehensive review of 51 studies from 14 countries found that while low-cost optical sensors have expanded air-quality monitoring capacity, their performance remains highly dependent on humidity, aerosol type, and calibration strategy [14].

Building on these broader evaluations, previous studies have assessed specific low-cost PM sensors—including the Sensirion SPS30—using standard laboratory aerosols such as sodium chloride (NaCl), Arizona Road Dust (ARD), and polyalphaolefin (PAO) oil droplets [11]. These studies have provided important insights into linearity, bias, and inter-sensor variability but offer limited evidence on performance in the complex aerosol mixtures typical of indoor environments. Only a small number of studies have examined emissions from candles [15], one of the most common sources of indoor fine particles, and evaluations involving cooking-generated aerosols remain notably scarce despite their central role in indoor exposure [14]. Available chamber studies indicate that low-cost sensors can detect cooking events but frequently underestimate mass concentrations, particularly for ultrafine-rich emissions [16]. Even in field settings, sensors performing well outdoors may miss or mischaracterize indoor exposure peaks arising from routine household activities [17].

Thus, while the validation literature has expanded, realistic indoor emission sources, including cooking, candles, and cigarette smoke, remain underrepresented. This gap limits the interpretability of low-cost sensor data in environments where personal exposure predominantly occurs.

This article introduces and evaluates ARMIE (ARhythmia, Myocardial Infarction & the Environment), a portable air quality monitor integrating the SPS30 optical particle PM sensor (Sensirion AG, Stäfa, Switzerland) alongside communication, power, and data-logging components.

The objective of the study was to (1) characterize the ARMIE nodes’ technical design and detection principles, (2) assess the accuracy of raw PM_2.5_ readings relative to a co-located comparison and higher-grade instruments across different emission sources and environmental conditions, using controlled chamber experiments with aerosols from cooking, candles, and cigarette smoke, and (3) evaluate calibration strategies to improve measurement agreement under varying conditions.

## 2. Materials and Methods

### 2.1. Instruments

Three types of instruments were used in this study: a low-cost PM sensor (ARMIE), a mid-cost optical instrument, and advanced reference monitors. Each instrument type is described in more detail below. This collocated approach, which utilizes advanced reference monitors for determining particle mobility and aerodynamic size distributions, along with an established standard optical instrument for mass estimation, aligns with established evaluation principles described in the Environmental Protection Agency (EPA) Air Sensor Toolbox [18] and other recent sensor evaluation frameworks.

#### 2.1.1. Low-Cost Sensor Node: The ARMIE Node

The ARMIE node is a portable air-monitoring device assembled by the Department of Public Health, Aarhus University, Denmark, in collaboration with Devlabs ApS, specifically designed for personal exposure monitoring studies. The unit measures 14.5 cm × 8.5 cm × 3.5 cm (height × width × depth), weighs approximately 275 g, and is designed to be worn with a shoulder strap that positions the sensor near the hip during use. A picture of the ARMIE node is found in Appendix D, Figure A9. The ARMIE node incorporates a Sensirion SPS30 optical particulate matter sensor to provide real-time PM_2.5_ measurements. Because optical PM sensors have well-documented limitations in accurately detecting coarse-mode particles (PM_10_), as they cannot reliably size them and they can be lost through settling and impaction, this study focuses exclusively on PM_2.5_ [19,20]. Additional environmental and contextual data are captured by the Sensirion SHTC3 sensor (temperature and relative humidity (RH)). A Quectel BG86 module enables cellular and wireless connectivity for geolocation (GPS) and data transmission. The ARMIE node records data at a five-minute interval, with each logged value representing the one-minute average from an active sampling period conducted within that interval. In line with its intended application, the ARMIE node is designed primarily for assessing indoor personal exposure patterns, where pollutant levels are strongly influenced by individual behaviors and microenvironmental conditions. An overview of all the individual sensors incorporated in the ARMIE node is listed in Appendix A, Table A1.

#### 2.1.2. Mid-Cost Instrument: DustTrak

A TSI DustTrak DRX (Model 8533, TSI Inc., Shoreview, MN, USA) was included as a mid-cost, optical real-time aerosol monitor. While not a regulatory reference instrument, DustTrak is widely used in field validation studies of low-cost sensors. Its inclusion ensured methodological consistency with prior literature and provided an additional comparator that bridges laboratory-grade reference instruments with real-time, field-deployable measurements. DustTrak logged 1-min mean PM_2.5_ measurements.

#### 2.1.3. Reference Instruments: Scanning Mobility Particle Sizer

A TSI Scanning Mobility Particle Sizer (SMPS), with a Differential Mobility Analyser (DMA, model 3081, TSI Inc., Minnesota, USA) and a Condensation Particle Counter (CPC, model 3775, TSI Inc., Minnesota, USA), was used to characterize submicron particles based on electrical mobility. The SMPS was set to measure particles in the range 0.015 to 0.66 µm; however, to ensure comparability with the optical instruments, data < 0.3 µm were excluded from further analyses. For larger particles, a TSI Aerodynamic Particle Sizer (APS, Model 3321, TSI Inc., Minnesota, USA) provided aerodynamic diameters from 0.8 to 20 µm. SMPS scan time was set to 125 s, and the APS was sampled at an interval of 20 s. Data from the SMPS and APS were extracted using the TSI Aerosol Instrument Manager software (AIM, version 10.3.1.0). Particles larger than 2.5 µm were removed in the analysis when comparing with the PM_2.5_ size fraction.

Size-resolved number concentrations from the SMPS (0.3–0.66 µm) and APS (0.8–2.5 µm) were converted to mass concentrations under a unit-density, spherical-particle assumption. For the SMPS, mass distributions were exported directly from TSI Aerosol Instrument Manager using a particle density of 1.0 g cm^3^ and a dynamic shape factor χ = 1.0. For the APS, mass concentrations were calculated from the number size distributions by applying the same density (1.0 g cm^3^) and spherical-particle assumption. These conversions yield a quasi-PM_2.5_ mass metric that is suitable for relative comparison with the optical instruments, but they do not account for source-specific differences in particle composition or effective density.

Together, the SMPS and APS provided reference data covering 0.3–0.66 µm (SMPS) and 0.8–2.5 µm (APS), leaving a narrow gap between 0.66 and 0.8 µm where neither instrument operated. The choice of SMPS and APS reflects their status as widely accepted reference technologies in aerosol science. The SMPS is considered the most accurate instrument for submicron size distributions, while the APS provides aerodynamic diameters that are directly relevant for inhalation health and PM mass metrics.

### 2.2. Experimental Setup

Experiments were conducted in a climate-controlled chamber located at the Aerosol and Climate Laboratory, Lund University, Sweden. The chamber had internal surfaces of stainless steel to minimize electrostatic deposition and emissions from volatile materials. It had a total volume of approximately 24 m^3^ (2.4 m × 3.6 m × 2.85 m) plus a ventilated airlock to avoid contamination from the outside. It could operate at temperatures from 5 °C to 50 °C and RH levels between 20% and 80%. Ventilation rates were adjustable, with inlet airflow ranging from 0 to 80 L/s or 0–12 air changes per hour (ACH). The chamber is equipped with integrated measurements for temperature, RH, and airflow, and these parameters are continuously displayed on a digital monitoring panel. A ceiling-mounted climate fan provides continuous air mixing and supports the stabilization of temperature and RH during experiments. The incoming air is conditioned through a multi-stage filtration and treatment system, including a pre-filter, a carbon filter, and a High-Efficiency Particulate Air (HEPA) filter, as well as a pre-heater and an acoustic silencer. Both the SMPS and APS were equipped with silica dryers during high-humidity experiments. Although volumetric flow rates were controlled, airflow velocities inside the chamber were not measured directly, and therefore, Reynolds numbers could not be determined. However, the chamber flow regime was designed to maintain well-mixed conditions.

### 2.3. Aerosol Generation

Aerosols were generated within the chamber using candles, cooking activities, and cigarette smoke (details for each source are provided below). The ARMIE node is intended for personal exposure monitoring within a cohort of patients with implantable cardioverter-defibrillators (ICDs). Therefore, cigarette smoke was also included as an evaluation source. Tobacco combustion is a frequent indoor emission and a major risk factor for cardiac events in this patient group. To ensure homogeneous particle distribution, a fan was placed inside the chamber to facilitate air mixing. All aerosol generation occurred entirely within the enclosed chamber under controlled conditions. A schematic of the chamber showing the location of the instruments and fan is provided in Appendix D.

#### 2.3.1. Clean Air

For clean air measurements, the chamber was flushed with filtered air. Filtration for clean air measurements utilized the pre-, carbon, and HEPA filter system.

#### 2.3.2. Candles

Four candles were lit, extinguished, and then relit, after which they remained burning for 10 min. Throughout the experiment, the environmental chamber was maintained at 20 °C, 40% RH, and an air flow rate of 5 L/s (0.75 ACH). The procedure was repeated under high-humidity conditions at 20 °C with RH levels of 70–80%.

#### 2.3.3. Cooking

Cooking-generated particles were generated by preparing pancakes using a Trangia gas burner and an IKEA non-stick pan, with pancake batter prepared according to a standard recipe. The environmental chamber was maintained at 20 °C, 40% RH, and an air flow rate of 5 L/s. During the 10-min cooking period, the airflow rate was increased to 20 L/s^1^ (3 ACH) to maintain acceptable CO_2_ concentrations inside the chamber while laboratory personnel were present to generate the aerosols. This procedure was repeated at 20 °C under high-humidity conditions (70–80% RH).

#### 2.3.4. Cigarettes

A Prince Rounded Taste cigarette was lit to generate aerosols inside the chamber. To prevent self-extinguishing, the cigarette was connected to a flow rate equal to 3 L/min that provided a brief suction pulse (approximately 3 s) every minute to maintain combustion. This procedure produced a stable side-stream aerosol without imposing a formal ISO puffing regime. Aerosol generation was conducted at 20 °C and 40% relative humidity, with the chamber ventilation rate increased to 30 L/s^1^ (4.5 ACH) to prevent odor accumulation. The experiment was not conducted under high-humidity conditions due to odor constraints.

### 2.4. Data Processing and Calibration Methods

To enable direct comparison across all instrument types, all data streams were synchronized using time stamps rounded to the nearest five-minute interval. All data were aggregated to five-minute means to match the sampling resolution of the low-cost sensors. For each five-minute interval, the arithmetic mean of all comparison instruments (for PM_2.5_ and relative humidity) was calculated.

#### 2.4.1. Performance Evaluation Metrics

For the evaluation of the ARMIE node performance, we focused on four evaluation metrics: the Pearson linear correlation coefficient (*r*), relative percent error, and mean error with corresponding upper and lower limits of agreement (LoA). These metrics were selected because they collectively capture key aspects of sensor performance relevant for calibration and validation. Tables in Appendix A are expanded to include mean absolute error (MAE) and root mean squared error (RMSE).

The correlation coefficient quantifies the strength of the linear association between the sensor readings and the reference instrument, reflecting the sensor’s ability to reproduce the temporal variability in particle concentrations. Relative percent error indicates the proportional error deviation between the sensor and reference measurements. Mean error and the limits of agreement (LoA) describe systematic and random deviations, respectively, providing information on both accuracy and precision across the full measurement range.

The evaluation was not designed to assess regulatory compliance with EPA or OSHA performance standards. However, the co-location methodology, use of reference-grade instruments for mass, and comparison of raw and calibrated sensor outputs follow principles consistent with current EPA Air Sensor Toolbox guidance for evaluating the performance of low-cost particle sensors [18].

#### 2.4.2. Calibration Models

For each comparison instrument, two models were evaluated. The first model evaluated the raw PM_2.5_ readings from the ARMIE node. The second model evaluated corrected ARMIE readings corrected with a multiple linear regression model that incorporated the PM_2.5_ readings and RH readings from the ARMIE node as independent variables.

The multiple linear regression calibration model used the following formula:$${PM}_{2.5,\;\;\;ref\;}=a_{0}+a_{1}\times ARMIE\;{PM}_{2.5}+a_{2}\times RH+a_{3}\times ({ARMIE\;PM}_{2.5}\times RH)$$
where the intercept ($a_{0}$), sensitivity ($a_{1}$), and the correction coefficient ($a_{2}$ and $a_{3}$) are determined separately for each sensor in the leave-one-sensor out approach and for each source in the leave-one-source out approach.

Model performance was evaluated using two different cross-validation strategies. In the leave-one-sensor-out approach, data from one sensor node at a time were excluded during model training and subsequently used for independent testing. In the leave-one-source-out approach, all data corresponding to a specific exposure source (e.g., cooking, candles, cigarettes, or clean air) were withheld during model training, and the model was evaluated on the source that was left out.

Personal deployments of low-cost PM sensors are expected to occur across heterogeneous and partially unobserved indoor microenvironments, introducing variability that cannot be fully captured by environment-specific calibration models. Although more flexible or non-linear calibration approaches may improve fit within a given dataset, such models risk overfitting to source- or setting-specific characteristics and may perform poorly when applied across individuals and homes. Accordingly, we applied parsimonious parametric calibration models prioritizing interpretability and robustness across conditions.

#### 2.4.3. Sensitivity Analyses

We conducted a sensitivity analysis with the same approach as the main analysis. This analysis only included data if ARMIE node PM_2.5_ readings were below 50 µg/m^3^ and above 5 µg/m^3^. All tables and figures for this analysis are presented in Appendix B.

## 3. Results

### 3.1. Data Overview

Figure 1 illustrates PM_2.5_ mass concentrations over time for each aerosol source generated in the climate-controlled chamber.

As found in Table 1, particle concentrations varied widely by source. Candle burning produced the highest particle levels, with median mass concentrations of 110 µg/m^3^ (SMPS + APS) and 59 µg/m^3^ (DustTrak). Cooking emissions reached median values around 57 µg/m^3^, while cigarette smoke and high-RH cooking generated substantially lower levels (<30 µg/m^3^). Number concentrations followed the same pattern, highest for candles and lowest for cigarette smoke. Overall, candles generated the most intense and variable aerosol loads, with humidity consistently reducing measured mass across all sources.

An overview of the measurement time per sensor and source type is provided in Appendix A, Table A2. Data completeness was below 100% for all sensors, primarily due to intermittent connectivity losses.

### 3.2. Leave-One-Sensor-Out

The performance evaluation metrics from the leave-one-sensor-out calibration are presented in Table 2. All Metrics are generated after excluding all observations where comparison instrument PM_2.5_ readings < 5 µg/m^3^. Thereby, these metrics do not include any data from the clean air.

Across all ARMIE nodes, performance was highly consistent when evaluated against the DustTrak but considerably weaker when compared with the SMPS + APS. When benchmarked against the DustTrak, the ARMIE node showed uniformly high correlations (*r* = 0.95), confirming strong linearity. Calibration reduced the mean error from 9.25 µg/m^3^ to −1.11 µg/m^3^ and slightly narrowed the limits of agreement (from −30.45 to 48.95 µg/m^3^ uncalibrated to −34.08 to 31.86 µg/m^3^ calibrated). However, the mean relative error increased from 340% to 527% following calibration.

In comparison with the SMPS + APS, the ARMIE node showed weaker agreement overall. Correlations were low (*r* = 0.51−0.52), and mean errors ranged from −6.22 to −12.72 µg/m^3^, with broad limits of agreement (−202.6 to +177.16 µg/m^3^). Calibration produced only minor improvements: slight increases in correlation, modest reductions in mean error, and less skewed LoA. However, the mean relative error increased from 53% to 73% after calibration.

For an expanded table of Table 1, including MAE and RMSE based on parameters for each of the 9 sensors, see Appendix A, Table A3.

Figure 2 presents Bland-Altman plots showing the difference between ARMIE and reference PM_2.5_ measurements across all observations. Each point is color-coded by source. The orange dashed line indicates the mean error, with 95% limits of agreement shown as dotted orange lines. A linear regression line (black) summarizes the trend in measurement error across the concentration range. Calibration produced a visible shift in the error trend for the DustTrak comparison: from a positive slope in the uncalibrated data to a negative slope after calibration, indicating a correction of systematic overestimation. Appendix A, Figure A1 shows corresponding scatterplots faceted by emission source.

Figure 2 shows heteroscedasticity across comparison instruments as well as calibrated and uncalibrated data. Figure 2 shows distinct source-dependent patterns in the residuals. Candle emissions, which are dominated by soot aggregates with low scattering efficiency, tend to be underestimated by the ARMIE node when compared with the SMPS + APS reference. In contrast, aerosols produced by cooking and cigarette smoke consist mainly of organic particles generated at lower combustion temperatures, which exhibit stronger light-scattering properties. These aerosols therefore lead to an overestimation by the ARMIE node relative to the SMPS + APS reference.

Relative errors were large for both the DustTrak and SMPS + APS comparisons. Figure 3 provides a graphical inspection of this pattern, showing that the most extreme relative errors originate primarily from observations with ARMIE PM_2.5_ readings ≤ 5 µg/m^3^. When stratified by concentration range, relative error distributions were markedly inflated in the low-concentration group, regardless of calibration status or reference instrument. At higher concentrations (>5 µg/m^3^), relative errors were substantially lower. For source-specific relative errors across individual observations, see Appendix A, Figure A2.

Figure 3 shows a stable relative error when the ARMIE readings < 5 ug/m^3^ are excluded. A scatterplot with relative error on the Y-axis and the reference PM_2.5_ on the x-axis is found in Appendix A Figure A2.

### 3.3. Leave-One-Source-Out

The performance evaluation metrics from the leave-one-source-out calibration are presented in Table 2. All Metrics are generated after excluding all observations where comparison instrument PM_2.5_ readings < 5 µg/m. Thereby, these metrics do not include any data from the clean air. Across all emission sources, the ARMIE node maintain consistently strong agreement compared to the DustTrak, while larger discrepancies were observed when compared to the SMPS + APS. For the DustTrak, correlations remained high (*r* = 0.93–0.98) across all aerosol types, confirming stable linearity of response under variable emission conditions. Calibration generally reduced systematic bias, also under elevated relative humidity, shifting most mean errors toward zero and tightening the limits of agreement, albeit not for the candles source.

In contrast, comparisons with the SMPS + APS showed moderate correlations (*r* = 0.78–0.92) and consistently higher mean errors than those observed for the DustTrak. Even after calibration, mean errors remained substantial (changing from −96.24 to −130.83 µg/m^3^ for candles and from 62.33 to 160.89 µg/m^3^ for cooking), and the span between lower and upper limits of agreement ranged broadly between −408 and 368 µg/m. All performance metrics comparing the ARMIE node with DustTrak are presented in Table 3, whereas comparisons with SMPS + APS are presented in Table 4. For MAE and RMSE, please refer to Appendix A Table A4.

Figure 4 presents Bland-Altman plots stratified by emission source, comparing ARMIE PM_2.5_ readings with the DustTrak and SMPS + APS across calibration states. Each point represents a single observation, and the panels show how measurement error varies by source and reference instrument. Orange and blue dashed lines denote the mean error and 95% limits of agreement for the DustTrak and SMPS + APS, respectively. A linear regression line is included to illustrate the trend in measurement error across the concentration range.

Comparison with DustTrak revealed source-specific trends in the slope of the error line. For candle emissions, calibration increased the negative slope, while for candles under high relative humidity, the slope shifted from positive to negative—indicating a correction of systematic overestimation. For cooking and cooking under high RH, the slope visibly flattened, suggesting reduced proportional bias after calibration. In contrast, for cigarette smoke, the slope remained largely unchanged. When compared to the SMPS + APS, the slope after calibration steepened for most sources, reflecting persistent over- and underestimation at higher concentrations.

Figure 4 shows heteroscedasticity across comparison instruments, aerosol sources, as well as calibrated and uncalibrated data.

## 4. Discussion

### 4.1. Overall Sensor Performance and Calibration Outcomes

The validation experiments showed that ARMIE nodes tracked the DustTrak reference with high correlation (*r* > 0.93) across all emission sources, consistent with earlier laboratory studies [11,12]. This performance pattern aligns with previous SPS30 evaluations conducted in indoor or residential environments, where strong linear agreement with optical reference instruments has also been observed [11,12,21].

Correlation with SMPS + APS was weaker (*r* = 0.51–0.92), likely due to differences in detection principles and the complexity of real-world aerosols such as cooking or cigarette smoke. Unlike the ARMIE node and DustTrak, which both rely on light scattering, the SMPS and APS generate number-based size distributions using distinct classification methods, limiting direct comparability. Prior evaluations have similarly reported that SPS30 performance deteriorates when compared against mobility- or aerodynamic-based reference system [13].

Leave-one-source-out analyses confirmed that calibration performance declined when a source was withheld from training. Against the DustTrak, calibration reduced mean error, but LoA remained similar. Against SMPS + APS, calibration effects were inconsistent, with several sources showing increased error and broader LoA, suggesting limited generalizability across particle types.

In contrast, the leave-one-sensor-out approach, which included all sources during training, yielded improved performance across most metrics, particularly when using the DustTrak. Mean relative error remained high in both setups due to instability at low concentrations.

A sensitivity analysis restricted to ARMIE PM_2.5_ readings between 50 and 5 µg/m^3^ showed improved agreement when calibrated with DustTrak (mean error −5.26 µg/m^3^; LoA −15.38 to 4.86 µg/m^3^), indicating that the ARMIE node performs best in the moderate exposure range relevant to epidemiological research. Finally, strong inter-sensor consistency supports their deployment in larger networks, although modest variation in calibration coefficients underscores the value of batch-wise correction or harmonization procedures.

### 4.2. Exposure Assessment in Environmental Epidemiology

In the context of particulate matter exposure assessment, a high correlation between low-cost sensors and reference instruments does not imply agreement in absolute concentrations. As Altman and Bland [22] argued, correlation captures linear strength of the association, not equivalence, and can be misleading when used in isolation. For epidemiological applications, the critical concern is whether measured values reflect true exposures with sufficient accuracy to inform individual risk classification. Limits of agreement (LoA) provide a more appropriate basis for this assessment, as they define the interval within which 95% of individual differences are expected to lie [23,24]. For example, the LoA between the calibrated ARMIE and the DustTrak (−34.04 to 31.86 µg/m^3^) implies that a sensor must report at least 49.04 µg/m^3^ before one can be 97.5% confident that the DustTrak readings of PM_2.5_ concentration exceed the WHO-recommended daily average of 15 µg/m^3^ [25]. Conversely, a sensor reading below 15 µg/m^3^ does not guarantee that DustTrak readings of PM_2.5_ concentration are below this threshold; achieving 97.5% confidence would require a physically implausible negative reading of −17.2 µg/m^3^ (15–31.86). In a sensitivity analysis restricted to moderate concentrations (ARMIE PM_2.5_ readings between 5 and 50 µg/m^3^), the mean error was −0.43 µg/m^3^ for uncalibrated and −5.26 µg/m^3^ for calibrated sensors, with LoA of −10.02 to 9.16 µg/m^3^ and −15.38 to 4.86 µg/m^3^, respectively. This implies that a calibrated ARMIE node would need to report at least 30.18 µg/m^3^ (15 + 15.18) before one can be 97.5% confident that the DustTrak readings of PM_2.5_ concentration exceed the WHO guideline of 15 µg/m^3^.

Such high assurance thresholds are not rare. As shown by Omelekhina et al. [6], indoor PM_2.5_ concentrations above 30 µg/m^3^ are frequently observed during everyday activities such as cooking, burning candles, and using e-cigarettes.

Moreover, heteroscedasticity was evident in the Bland-Altman plots, with measurement error increasing at higher concentrations. Since the calculation of LoA assumes constant variance across the measurement range, this violation limits its interpretability, particularly at the upper end of the distribution, where classification error may be greatest.

### 4.3. Hygroscopic Growth

Low-cost optical particle sensors are strongly influenced by relative humidity, as water can be absorbed by particles and lead to hygroscopic growth. This process often leads to an overestimation of PM_2.5_ mass concentrations [17].

In the ARMIE node, this limitation may be amplified because the SHTC3 sensor that measures temperature and RH is mounted inside the printed circuit board enclosure rather than sampling the inlet air directly. Consequently, the measured RH may not be representative of the true ambient RH, leading to a misclassification between the conditions driving hygroscopic particle growth and those used for calibration.

Previous studies suggest that the SPS30 sensor implemented in the ARMIE node is less affected by RH than several other low-cost optical particle sensors, though overestimation remains a relevant concern, particularly above 70–80% RH [11,12]. In our controlled chamber experiments, RH was set to target levels of 40% and 80%, but substantial fluctuations occurred, particularly during cooking events and under high-humidity scenarios. This variability highlights the practical challenge of maintaining stable RH conditions in source-specific experiments and underscores the importance of incorporating humidity adjustment strategies into calibration models. Calibration approaches addressing hygroscopic growth have been examined in previous studies. Ref. [26] reported that optical particle counter measurements under RH conditions above 60% should be corrected, and [27] similarly found RH to influence measurement bias. For the intended use of the ARMIE node, the target population is expected to spend more than 90% of the monitoring period indoors, where RH typically remains below these thresholds. Consequently, RH-related corrections would have minimal impact on the overall exposure estimates.

### 4.4. Reproducibility

The controlled chamber experiments revealed notable variation in the peak PM_2.5_ concentrations between repeated trials using the same emission source. This variability underscores that absolute reproducibility of concentration profiles is difficult to achieve, even under carefully standardized laboratory conditions. Importantly, this mirrors the inherent heterogeneity of real-world indoor exposures. Cooking events, for instance, generate emissions that vary substantially with food type, cooking oil, and ventilation, leading to order-of-magnitude differences in particle number and mass concentrations [28]. Similarly, comprehensive in situ work in Swedish residences has shown that indoor particle levels are strongly dominated by variable indoor sources such as cooking, candle burning, and e-cigarette use, with indoor concentrations often exceeding outdoor levels by factors of five or more [6].

In this context, the modest variability observed between nominally identical chamber trials should not be regarded as a limitation but rather as a proxy for real-world variability. Real-world household exposures are influenced by factors as diverse as the composition of a candle wick, the use of an exhaust hood, or the style of meal preparation. These sources of stochasticity mean that personal exposure can rarely be reduced to a single “true” emission profile. For sensor validation aimed at personal exposure monitoring, the central objective is therefore not to reproduce identical concentration peaks, but to evaluate sensor performance under conditions representative of the environments in which participants are exposed.

### 4.5. Temporal Resolution and Data Alignment

A key limitation of this evaluation is the mismatch in temporal resolution between the ARMIE nodes and the higher-grade and reference instruments. The ARMIE node reports one value every five minutes, whereas SMPS reports every 125 s, the DustTrak every minute, and the APS every ten seconds. For analysis, all data were averaged to match the five-minute resolution of the ARMIE node. However, the ARMIE node does not measure continuously within each interval; it samples for one minute every five minutes, and the units were not synchronized to measure at the same time. This unavoidably smoothed sharp peaks and occasionally shifted their timing. Similar challenges with averaging and synchronization have been described in other evaluations of low-cost sensors [21].

### 4.6. Challenges in Comparing PM_2.5_ Across Instruments

A major challenge in this evaluation stems from the differing physical principles each instrument applies when classifying particle size. The SMPS reports mobility diameters, the APS aerodynamic diameters, while both the ARMIE node and DustTrak report optical diameters. These size definitions are not directly comparable, as they depend differently on particle morphology, density, and refractive index. Converting between them requires assumptions about particle shape and density that can shift derived mass concentrations substantially—in some cases by a factor of three to five, especially in the accumulation and coarse modes.

Ref. [29] demonstrated these discrepancies in a direct comparison of SMPS, APS, fast mobility particle sizers (FMPS), and electrical low-pressure impactors (ELPI). While reasonable agreement was observed in the mid-size range, substantial divergence occurred at both the ultrafine (<50 nm) and coarse (>1 µm) ends of the distribution. In the present study, such definitional mismatches likely contributed to the weaker agreement between the ARMIE node and the combined SMPS + APS reference. By contrast, DustTrak shares an optical measurement principle with the ARMIE node, allowing for more coherent mass estimates without the need to reconcile fundamentally different size metrics.

These limitations underscore that the reference PM_2.5_ derived from merged size distributions is not absolute, but highly conditional on conversion assumptions. These discrepancies are also visible in Figure 1, where the degree of divergence between the DustTrak and the SMPS + APS varies across emission sources, reflecting the different physical measurement principles of the instruments and the source-specific composition and size distribution of the generated aerosols. As such, evaluation studies relying on hybrid SMPS + APS systems must account for this uncertainty when benchmarking optical sensors. This is consistent with previous findings that even reference-grade optical instruments such as the DustTrak can show substantial deviations under varying aerosol types unless specifically calibrated [30], and that strong correlation in temporal trends does not imply agreement in absolute mass concentration [31].

### 4.7. Assumptions in SMPS + APS Number to Mass Conversion

An important methodological limitation is that the particle mass concentrations derived from the SMPS and APS relied on simplified number–to–mass conversions using a unit density (1.0 g cm^−3^) and a spherical particle shape factor. These assumptions do not capture the true effective densities of the source aerosols and therefore introduce uncertainty in the absolute mass estimates.

### 4.8. Absence of Gravimetric Reference Measurements

A further limitation is that no filter-based gravimetric measurements were collected during the chamber experiments. As a result, the mass concentrations derived from the SMPS and APS cannot be corrected to a traceable gravimetric standard. This restricts the interpretation of the SMPS + APS data to relative comparisons rather than absolute mass validation.

### 4.9. Sampling Frequency

Another limitation of this evaluation was inconsistent data uploading from the ARMIE nodes, resulting in variation in the number of observations across sources and devices. For example, during the candle experiments, most sensors provided 36–39 observations, while sensor 196 only logged 14, and sensor 199 logged 15. Similar gaps were observed across other sources, with the problem most pronounced in the cooking experiments, where some sensors recorded as few as 16 observations. These irregularities reduce comparability between sensors and may introduce bias if performance metrics are calculated from unequal sample sizes. The chamber setup itself may have contributed, as the experiments were conducted in steel chambers with weakened signal transmission, which increased the risk of failed uploads. For a full overview of data completeness, see Appendix A, Table A2.

### 4.10. Perspectives

Epidemiologic studies have long relied on fixed-site monitors as proxies for personal exposure. Yet, these ambient data often misrepresent the concentrations to which participants are exposed, particularly when indoor emissions dominate daily life. While such exposure misclassification is well known, it has not prevented the discovery of robust health associations.

Our findings show that low-cost sensors such as the ARMIE node perform with high correlation when compared with mid-cost instruments like the DustTrak. This convergence signals a methodological shift, where personal exposure monitoring with validated, low-cost devices has become feasible. For several epidemiological questions, particularly those relying on within-person contrasts over time, the ability to capture exposure variability and timing may be as important as accuracy with a reference instrument. Rather than striving only for accuracy, emphasis should now turn to representativeness, capturing how everyday environments drive the variability in personal exposure. With low-cost sensors, Public Health research can focus on how personal fluctuations in PM_2.5_ may impact the risk of disease.

## 5. Conclusions

In this study, we demonstrated that the ARMIE nodes exhibit strong linear correlations with an optical reference instrument (DustTrak) across a range of realistic indoor emission sources, consistent with previous evaluations of low-cost optical PM_2.5_ sensors. However, agreement analysis demonstrated that correlation alone is insufficient for characterizing measurement accuracy in exposure assessment. Bland-Altman analyses showed substantial disagreement at the observation level, including concentration-dependent error and heteroscedasticity, with increasing error at increasing PM_2.5_ levels.

Using raw (uncalibrated) measurements, the ARMIE node tracked temporal variability well relative to the DustTrak (r ≈ 0.95) but showed some disagreement reflected in the mean error (9.25 µg/m^3^) and LoA (−30.45 to 48.95 µg/m^3^). In contrast, correlation with SMPS + APS was substantially weaker (r ≈ 0.51; mean error = −12.72 µg/m^3^; LoA = −202.60 to 177.16 µg/m^3^), reflecting differences in measurement principles and mass derivation assumptions.

A linear calibration model including relative humidity reduced mean error when compared with DustTrak (mean error −1.11 µg/m^3^) and modestly narrowed LoA (−34.08 to 31.85 µg/m^3^), but did not eliminate heteroscedasticity or the source-dependent discrepancies. Compared with SMPS + APS concentrations, calibration yielded only marginal changes in correlation and LoA and did not resolve the large source-dependent discrepancies, indicating limited transferability of a single global correction across particle composition and size distribution.

These findings underscore that calibration of low-cost PM sensors should be treated as source- and condition-dependent, and future deployments should prioritize (i) calibration strategies matched to expected aerosol mixtures and RH regimes, and (ii) analytical frameworks that propagate measurement uncertainty (e.g., LoA-informed thresholds or measurement-error models) rather than relying on correlation metrics alone.

Taken together, these results show that ARMIE node data are best interpreted as providing reliable information on relative within-person variation and exposure timing in indoor environments, while absolute mass concentrations remain conditional on aerosol type, humidity, and the chosen reference framework.

## Figures and Tables

**Figure 1 sensors-26-00280-f001:**
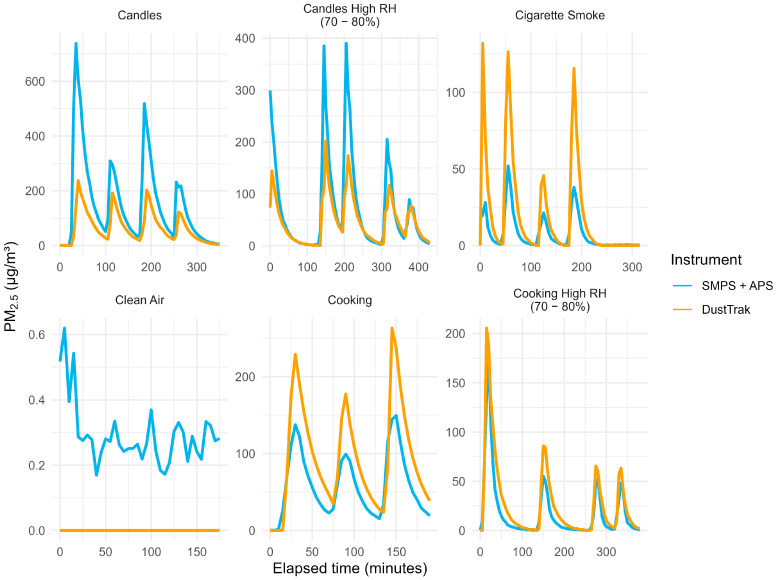
Time series of PM_2.5_ mass concentrations for each source. Time series of PM_2.5_ mass concentrations by emission source. DustTrak refers to measurements from the TSI DustTrak. SMPS + APS denotes PM_2.5_ mass concentrations derived from a TSI Scanning Mobility Particle Sizer (SMPS; 0.3–0.66 µm) combined with a TSI Aerodynamic Particle Sizer (APS; 0.8–2.5 µm). Note: axis scales vary between panels.

**Figure 2 sensors-26-00280-f002:**
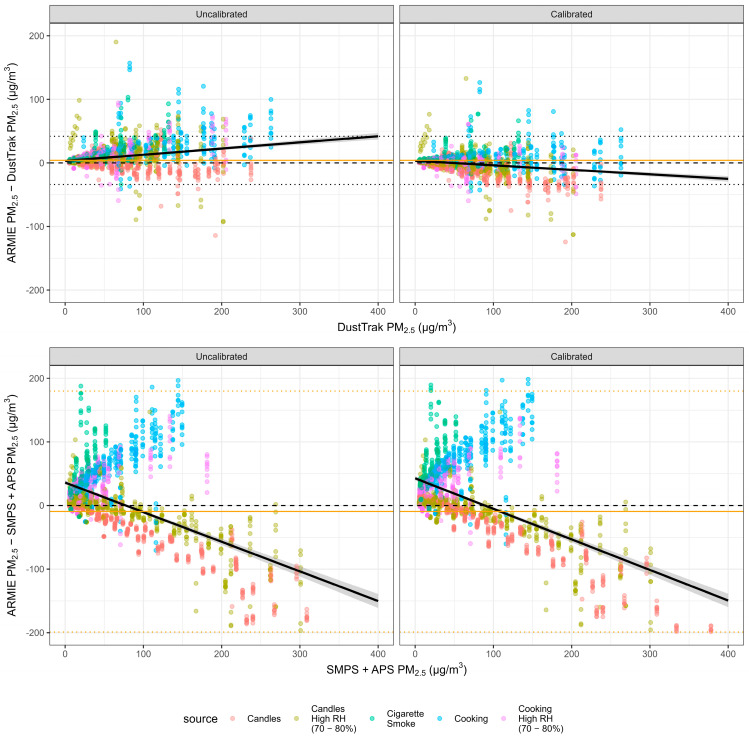
Bland-Altman plot based on the leave-one-sensor-out approach. Bland-Altman plots comparing ARMIE PM_2.5_ with reference instruments. Panels on the left show uncalibrated data; panels on the right show calibrated data. Calibrated values were obtained using a linear regression model including ARMIE PM_2.5_ and RH as predictors. Bold lines denote mean error, and dotted lines indicate the 95% Bland-Altman limits of agreement. Comparisons are shown separately for the TSI DustTrak and for PM_2.5_ derived from a TSI Scanning Mobility Particle Sizer (SMPS; 0.3–0.66 µm) combined with a TSI Aerodynamic Particle Sizer (APS; 0.8–2.5 µm). Note: 207 observations were excluded because their error values fell outside the displayed axis range.

**Figure 3 sensors-26-00280-f003:**
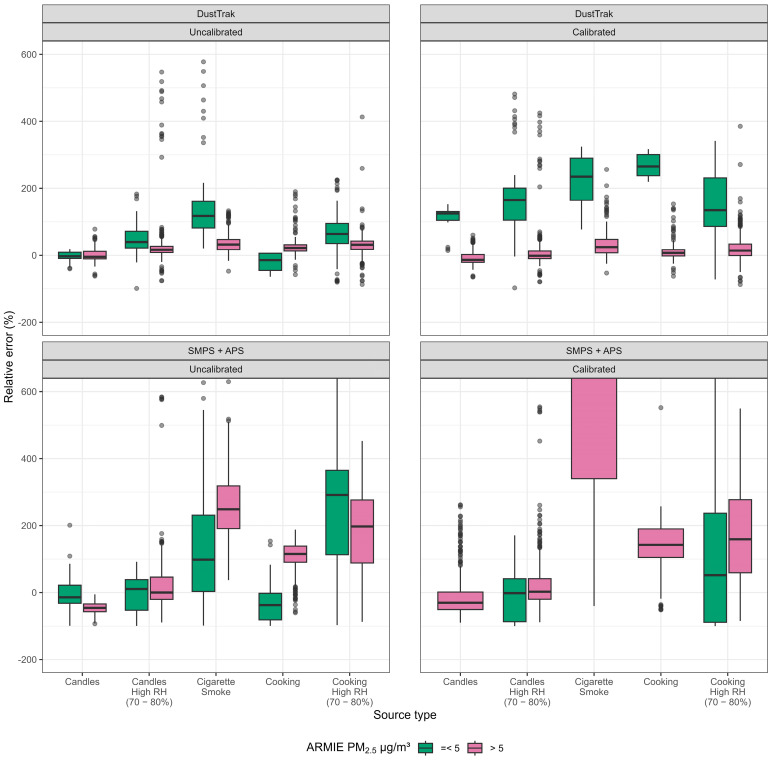
Distribution of relative error of ARMIE PM_2.5_ readings by source and calibration status. Box-and-whisker plots with dots showing relative error (%) of ARMIE PM_2.5_ by emission source and calibration status compared with reference instruments. Calibrated values were obtained using a linear regression model including ARMIE PM_2.5_ and RH as predictors. The top panels show comparisons with the TSI DustTrak, and the bottom panels show comparisons with PM_2.5_ derived from a TSI Scanning Mobility Particle Sizer (SMPS; 0.3–0.66 µm) combined with a TSI Aerodynamic Particle Sizer (APS; 0.8–2.5 µm). Boxplots are stratified by concentration range (PM_2.5_ ≤ 5 µg/m^3^ vs. >5 µg/m^3^). Note: 624 observations were excluded because their relative error values fell outside the displayed axis range.

**Figure 4 sensors-26-00280-f004:**
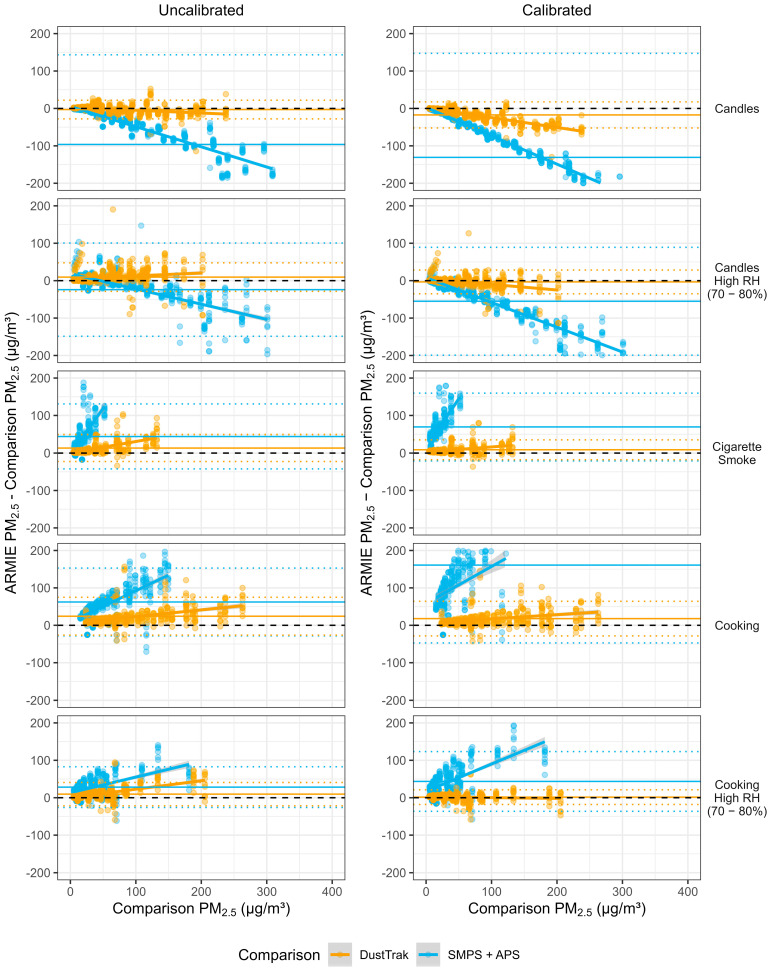
Bland-Altman plot based on the leave-one-source-out approach. Bland-Altman plots comparing ARMIE PM_2.5_ with PM_2.5_ derived from a TSI Scanning Mobility Particle Sizer (SMPS; 0.3–0.66 µm) combined with a TSI Aerodynamic Particle Sizer (APS; 0.8–2.5 µm). Panels on the left show uncalibrated data; panels on the right show calibrated data. Calibrated values were obtained using a linear regression model including ARMIE PM_2.5_ and RH as predictors. Bold lines denote mean error, and dotted lines indicate the 95% Bland-Altman limits of agreement. Note: 207 observations were excluded because their error values fell outside the displayed axis range.

**Table 1 sensors-26-00280-t001:** Descriptive statistics of particle counts and concentrations measured during chamber experiments stratified by aerosol sources and RH conditions.

Source	RH Condition	Instrument	Median	Mean	SD	Min	Max
Candles	Normal	SMPS + APS (#/cm^3^)	2188	3529	3414	149	16,339
		SMPS + APS (µg/m^3^)	110	170	161	7	739
		DustTrak (µg/m^3^)	59	73	57	5	238
	High RH (70–80%)	SMPS + APS (#/cm^3^)	1169	2008	2049	117	8774
		SMPS + APS (µg/m^3^)	49	92	96	5	390
		DustTrak (µg/m^3^)	48	58	45	6	202
Cigarette smoke	Normal	SMPS + APS (#/cm^3^)	306	383	246	101	863
		SMPS + APS (µg/m^3^)	20	21	12	6	52
		DustTrak (µg/m^3^)	48	58	35	17	132
Cooking	Normal	SMPS + APS (#/cm^3^)	1036	1223	739	303	3055
		SMPS + APS (µg/m^3^)	57	65	40	15	149
		DustTrak (µg/m^3^)	85	105	64	24	263
	High RH (70–80%)	SMPS + APS (#/cm^3^)	402	541	464	109	2070
		SMPS + APS (µg/m^3^)	26	37	38	6	181
		DustTrak (µg/m^3^)	44	56	45	11	206

PM_2.5_ concentrations derived from SMPS + APS were obtained by combining measurements from a TSI Scanning Mobility Particle Sizer (SMPS; 0.3–0.66 µm) and a TSI Aerodynamic Particle Sizer (APS; 0.8–2.5 µm). Observations with DustTrak or SMPS + APS reference PM_2.5_ < 5 µg/m^3^ were excluded before calculation of descriptive statistics. # denotes particle number concentration.

**Table 2 sensors-26-00280-t002:** Performance metrics for ARMIE PM_2.5_ readings compared with DustTrak and SMPS + APS using the leave-one-sensor-out validation approach.

Comparison Instrument	Method	Mean Relative Error (%)	Mean Error (µg/m^3^)	LoA Low (µg/m^3^)	LoA High (µg/m^3^)	*r*
DustTrak	Calibrated	6.12	−1.11	−34.08	31.86	0.95
	Uncalibrated	20.63	9.25	−30.45	48.95	0.95
SMPS + APS	Calibrated	72.86	−6.22	−195.19	182.74	0.52
	Uncalibrated	52.51	−12.72	−202.60	177.16	0.51

Metrics are aggregated across nine ARMIE nodes. Uncalibrated values refer to raw ARMIE PM_2.5_ measurements. Calibrated values were obtained using a linear regression model including ARMIE PM_2.5_ and RH as predictors. PM_2.5_ from SMPS + APS was derived by combining measurements from a TSI Scanning Mobility Particle Sizer (SMPS; 0.3–0.66 µm) and a TSI Aerodynamic Particle Sizer (APS; 0.8–2.5 µm). LoA denotes Bland-Altman limits of agreement. Observations with reference PM_2.5_ < 5 µg/m^3^ were excluded prior to metric calculation.

**Table 3 sensors-26-00280-t003:** Performance of ARMIE PM_2.5_ concentrations compared with DustTrak using a leave-one-source-out validation approach.

Source	RH Condition	Method	Mean Relative Error (%)	Mean Error (µg/m^3^)	LoA Low (µg/m^3^)	LoA High (µg/m^3^)	*r*
Candles	Normal	Calibrated	−20.39	−17.33	−52.24	17.58	0.98
		Uncalibrated	−0.18	−2.96	−27.89	21.98	0.98
	High RH (70–80%)	Calibrated	4.09	−3.32	−35.03	28.39	0.93
		Uncalibrated	26.61	9.49	−28.63	47.61	0.93
Cigarette smoke	Normal	Calibrated	32.09	8.86	−17.63	35.35	0.96
		Uncalibrated	33.46	13.42	−22.47	49.32	0.96
Cooking	Normal	Calibrated	19.46	17.91	−28.33	64.15	0.95
		Uncalibrated	25.48	24.26	−26.38	74.90	0.96
	High RH (70–80%)	Calibrated	15.54	1.58	−18.05	21.21	0.97
		Uncalibrated	25.50	9.57	−21.60	40.75	0.97

Metrics were generated using a leave-one-source-out approach. Uncalibrated values refer to raw ARMIE PM_2.5_ measurements. Calibrated values were obtained using a linear regression model including ARMIE PM_2.5_ and RH as predictors. LoA denotes Bland-Altman limits of agreement. Observations with reference PM_2.5_ < 5 µg/m^3^ were excluded before metric calculation.

**Table 4 sensors-26-00280-t004:** Performance of ARMIE PM_2.5_ concentrations compared with SMPS + APS using a leave-one-source-out validation approach.

Source	RH Condition	Method	Mean Relative Error (%)	Mean Error (µg/m^3^)	LoA Low (µg/m^3^)	LoA High (µg/m^3^)	*r*
Candles	Normal	Calibrated	−70.97	−130.83	−408.84	147.19	0.76
		Uncalibrated	−47.14	−96.24	−335.85	143.38	0.78
	High RH (70–80%)	Calibrated	−37.67	−55.02	−199.21	89.17	0.81
		Uncalibrated	16.16	−24.06	−148.66	100.54	0.78
Cigarette smoke	Normal	Calibrated	396.69	69.37	−20.93	159.67	0.78
		Uncalibrated	211.50	44.27	−42.34	130.88	0.78
Cooking	Normal	Calibrated	267.73	160.89	−47.09	368.86	0.90
		Uncalibrated	104.47	62.33	−28.27	152.92	0.92
	High RH (70–80%)	Calibrated	164.73	43.53	−36.30	123.37	0.91
		Uncalibrated	117.11	28.31	−26.12	82.75	0.92

PM_2.5_ concentrations from SMPS + APS were derived by combining measurements from a TSI Scanning Mobility Particle Sizer (SMPS; 0.3–0.66 µm) and a TSI Aerodynamic Particle Sizer (APS; 0.8–2.5 µm). Calibrated values were obtained using a linear regression model including ARMIE PM_2.5_ and RH as predictors. LoA denotes Bland-Altman limits of agreement. Observations with reference PM_2.5_ < 5 µg/m^3^ were excluded before metric calculation.

## Data Availability

The original data, as well as all code used to generate the results in this study, are openly available at the GitHub (Git version 2.50.1.windows.1) repository “ARMIELabEval” https://github.com/Kloppenborg/ARMIELabEval.git (accessed on 22 December 2025).

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
