# Peer review of "Laboratory Evaluation of ARMIE, a Portable SPS30-Based Low-Cost Sensor Node for PM2.5 Monitoring"

_sensors, 2026, doi:10.3390/s26010280_

Round 1

Reviewer 1 Report

Comments and Suggestions for Authors

General Comments
The manuscript evaluates a low-cost sensor (SPS30) against multiple aerosol types in a controlled exposure chamber. Although this work is not entirely novel, since the SPS30 has been evaluated previously in both laboratory and field settings, the experimental approach and selected aerosol types may still offer a meaningful contribution. I believe the manuscript is suitable for publication after the authors address the comments below.

Specific Comments

  1. Line 39: Please avoid using “particulate matter” and “aerosols” interchangeably. The phrase “particulate matter from aerosols” is redundant because PM represents the particulate portion of an aerosol. To avoid confusion, rephrase as “particles 2.5 μm in diameter and smaller.” Apply this correction throughout the manuscript.
  2. Line 40: The ARMIE should be clearly identified as an air quality monitor to distinguish it from the air quality sensor (SPS30). The SPS30 alone is not functional without connection to a computer or microcontroller. Please revise this consistently across the manuscript.
  3. Lines 75–82: Combine these sentences into a single paragraph and avoid starting with “We.” Instead, begin with “The objective of the study…”
  4. Lines 85–87: The first sentence should not appear in bold or in a different format. Please correct this throughout.
  5. Line 90: Spell out Environmental Protection Agency (EPA) and National Institute for Occupational Safety and Health (NIOSH).
  6. Line 90: Provide references for the EPA and NIOSH performance criteria.
  7. Line 92: Here the ARMIE is correctly referred to as a monitor. Please make this terminology consistent throughout.
  8. Line 104: Do not call this section “Appendix.” Use “Supplemental Material.”
  9. Line 104: Please include an actual photograph of the ARMIE monitor in the Supplemental Material.
  10. Line 126: Remove the colon from the sentence.
  11. Lines 113–128: Specify the densities and shape factors used for each aerosol type when converting number concentrations to mass concentrations for both the SMPS and APS.
  12. Lines 130–140: Add a schematic of the chamber that shows the locations of the instruments, the fan, and airflow direction.
  13. Lines 150–152: Merge this sentence with the previous paragraph for better flow.
  14. Line 155: Provide the Reynolds numbers for the flow associated with each aerosol. Turbulent flow can introduce bias and affect measurement accuracy. Also explain why different flows were used for different aerosol types.
  15. Line 166: State what smoking profile the aerosol generation followed.
  16. Lines 179–184: Indicate whether any EPA or OSHA criteria were followed when evaluating these results.
  17. Lines 193–196: The calibration description is unclear. Please revise and include the relevant equations to improve clarity.
  18. Line 206: Again, replace “Appendix” with “Supplemental Material.”
  19. Limitations: Add that filter-based gravimetric measurements were not collected, which limits the ability to correct SMPS and APS mass concentrations.
  20. Line 234: Confirm whether you meant Table 2.
  21. Table 2: The metrics appear to compare the SMPS, APS, and DustTrak. The ARMIE is not mentioned, even though it is the primary focus of the study. This is confusing and needs clarification.
  22. Figures 2 and 4: Do not use “SPS30” in the captions. Use “ARMIE” so that readers understand the SPS30 is inside the ARMIE. Maintain consistency across all figures and figure captions.
  23. Line 282: State clearly that Figure 3 contains box and whisker plots. Also clarify whether the data shown are from the ARMIE, because this is not clear.
  24. Discussion: Expand the discussion to include comparisons with similar work in the literature, especially studies using the SPS30 in residential or indoor settings.

Author Response

Reviewer 1:

Dear Reviewer,

Thank you very much for your careful evaluation of the manuscript. Before addressing your specific comments, we would like to offer one clarification regarding the calibration model used in the original submission. Following Reviewer 1’s remark about the missing calibration formula, we revisited the complete set of analysis functions and code. Through this review we identified that relative humidity (RH) had been implemented as an additive term rather than as an interaction with the ARMIE PM2.5 readings, which is the statistically appropriate specification given the underlying measurement behaviour.

We have therefore re-estimated all calibration models using an interaction term between ARMIE PM2.5 and RH. Although this represents an important methodological correction, its influence on the numerical results was limited. For the DustTrak-based calibration, the changes were confined to minor decimal-level differences. For the SMPS + APS calibration, the impact was more evident, but did not alter the interpretation of the results or the overall conclusions of the study.

All tables and figures have been updated accordingly, and the revised model specification is now clearly described in the manuscript.

We are grateful for the opportunity to correct this point and sincerely appreciate your contribution to improving the methodological clarity of the work.

Comments and Suggestions for Authors

General Comments
The manuscript evaluates a low-cost sensor (SPS30) against multiple aerosol types in a controlled exposure chamber. Although this work is not entirely novel, since the SPS30 has been evaluated previously in both laboratory and field settings, the experimental approach and selected aerosol types may still offer a meaningful contribution. I believe the manuscript is suitable for publication after the authors address the comments below.

Specific Comments

Line 39: Please avoid using “particulate matter” and “aerosols” interchangeably. The phrase “particulate matter from aerosols” is redundant because PM represents the particulate portion of an aerosol. To avoid confusion, rephrase as “particles 2.5 μm in diameter and smaller.” Apply this correction throughout the manuscript.

  • We have removed the redundant phrasing and revised the text [Line 39-40].

Line 40: The ARMIE should be clearly identified as an air quality monitor to distinguish it from the air quality sensor (SPS30). The SPS30 alone is not functional without connection to a computer or microcontroller. Please revise this consistently across the manuscript.

  • We appreciate this clarification. We have revised the text to consistently identify the ARMIE as an air quality monitor and the SPS30 as the integrated particle sensor. [line 84-85].

Lines 75–82: Combine these sentences into a single paragraph and avoid starting with “We.” Instead, begin with “The objective of the study…”

  • The sentences in [Lines 86-91] have been revised, and the section now begins with “The objective of the study…” rather than using the first-person formulation.

Lines 85–87: The first sentence should not appear in bold or in a different format. Please correct this throughout.

  • The affected sentence has been corrected, and the manuscript has been checked to ensure consistency with the typography and formatting specifications of the MDPI Word template.

Line 90: Spell out Environmental Protection Agency (EPA) and National Institute for Occupational Safety and Health (NIOSH).

  • We have revised the text [Line 99] to spell out “Environmental Protection Agency (EPA)” at first mention and ensured consistent abbreviation use throughout the manuscript.

Line 90: Provide references for the EPA and NIOSH performance criteria.

  • We agree that the original sentence suggested the use of specific EPA and NIOSH performance protocols, which was not the intention. We have therefore revised the text to avoid implying adherence to formal documents or criteria that were not explicitly applied. The updated wording now states that the collocated use of mobility-, aerodynamic- and optical-based reference instruments is consistent with established evaluation principles described in the United States Environmental Protection Agency (EPA) Air Sensor Toolbox and recent sensor evaluation frameworks [Line 98-100]. This phrasing more accurately reflects the methodological alignment without referencing specific protocols that do not exist as formal standards.

Line 92: Here the ARMIE is correctly referred to as a monitor. Please make this terminology consistent throughout.

  • This reference to the ARMIE as a monitor has now been applied consistently throughout the manuscript.

Line 104: Do not call this section “Appendix.” Use “Supplemental Material.”

  • We originally used the term “Supplementary Material,” but as the MDPI Sensors template specifies “Appendix” for additional material included within the manuscript file, we have retained this terminology to ensure consistency with the journal’s formatting requirements.

Line 104: Please include an actual photograph of the ARMIE monitor in the Supplemental Material.

  • We have added a photograph of the ARMIE air quality monitor to Appendix D (Figure D2), and highlighted this in the text [Line 106-107].

Line 126: Remove the colon from the sentence.

  • The colon has been removed, and the sentence has been revised.

Lines 113–128: Specify the densities and shape factors used for each aerosol type when converting number concentrations to mass concentrations for both the SMPS and APS.

  • Thank you for this helpful comment. In the original manuscript, the assumptions used for converting number to mass for the SMPS and APS were not described explicitly. We have now clarified that SMPS mass concentrations were exported from TSI Aerosol Instrument Manager using a unit density of 1.0 g cm⁻³ and a spherical-particle assumption (χ = 1.0), and that APS mass concentrations were calculated from the number size distributions using the same density and shape factor. These assumptions, which are commonly applied when composition-specific measurements are unavailable, have been added to the Methods section [Lines 139–147]. We also acknowledge in the limitations that using a single generic density across all aerosol types may introduce bias in absolute mass estimates, particularly for source-specific comparisons.

Lines 130–140: Add a schematic of the chamber that shows the locations of the instruments, the fan, and airflow direction.

  • We have added a schematic of the experimental chamber that illustrates the placement of the instruments, the mixing fan, and the airflow direction (Appendix D) [Line 186-188].

Lines 150–152: Merge this sentence with the previous paragraph for better flow.

  • We agree that the description of the clean-air condition must be clearly integrated into the experimental setup. To maintain consistency with the structure used for the other aerosol sources (candles, cooking emissions, cigarette smoke), we have retained “Clean air” as a separate subheading rather than merging it with the preceding paragraph [Line 185]. This ensures that each exposure condition is described using an identical format. The formatting and placement of this section have been revised for clarity.

Line 155: Provide the Reynolds numbers for the flow associated with each aerosol. Turbulent flow can introduce bias and affect measurement accuracy. Also explain why different flows were used for different aerosol types.

  • Thank you for raising this point. Airflow rates in the chamber were controlled during the experiments; however, airflow velocity was not measured directly. For this reason, Reynolds numbers cannot be reported. The flow settings were selected to achieve well-mixed conditions [line 170-173]. We briefly describe the reason to increase air flow under cooking and cigarette aerosol generations [Line 197-199 & 206-209]. We hope these are sufficient.

Line 166: State what smoking profile the aerosol generation followed.

  • We have clarified the smoking procedure used to generate the cigarette aerosol [Line 203-206].

Lines 179–184: Indicate whether any EPA or OSHA criteria were followed when evaluating these results.

  • The study was not designed to evaluate regulatory compliance with specific EPA or OSHA performance criteria, and no formal regulatory protocol was applied. We have clarified this in the manuscript. The evaluation approach - co-location with established reference instruments, assessment of linearity, agreement, and calibration - follows general principles described in EPA’s Air Sensor Toolbox for evaluating low-cost particulate sensors, but does not constitute a regulatory performance test [Line 230-234].

Lines 193–196: The calibration description is unclear. Please revise and include the relevant equations to improve clarity.

  • Thank you for this comment. We agree that the description of the calibration procedure required clarification, and we have revised the text accordingly [Line 245-249. As noted in our introductory statement, this comment prompted us to re-examine the underlying model specification, during which we identified and corrected the issue regarding the treatment of relative humidity. The relevant equations have now been added, and we sincerely thank the reviewer for helping us improve the clarity and correctness of this part of the manuscript.

Line 206: Again, replace “Appendix” with “Supplemental Material.”

  • In accordance with the MDPI Sensors template, additional material included within the manuscript is placed in an “Appendix”. We have therefore retained the term “Appendix” to remain consistent with the journal’s formatting requirements.

Limitations: Add that filter-based gravimetric measurements were not collected, which limits the ability to correct SMPS and APS mass concentrations.

  • We have added a dedicated subsection in the Limitations noting that no filter-based gravimetric measurements were collected during the chamber experiments. As a result, the SMPS + APS mass concentrations cannot be corrected to a traceable gravimetric reference and should therefore be interpreted as relative rather than absolute mass values. This limitation has now been explicitly acknowledged in the manuscript [Line 564-569].

Line 234: Confirm whether you meant Table 2.

  • Thank you for pointing out this incorrect number. The correct table is Table 2, just as you refer to. The table reference has been corrected.

Table 2: The metrics appear to compare the SMPS, APS, and DustTrak. The ARMIE is not mentioned, even though it is the primary focus of the study. This is confusing and needs clarification.

  • Thank you for pointing out the potential ambiguity in Table 2. We have revised the table title and caption to make it explicit that the performance metrics reflect comparisons between the ARMIE PM5 readings and the DustTrak and SMPS + APS reference instruments.

Figures 2 and 4: Do not use “SPS30” in the captions. Use “ARMIE” so that readers understand the SPS30 is inside the ARMIE. Maintain consistency across all figures and figure captions.

  • We have revised all figure captions and associated labels to use “ARMIE” rather than “SPS30,” ensuring consistency across figures and text.

Line 282: State clearly that Figure 3 contains box and whisker plots. Also clarify whether the data shown are from the ARMIE, because this is not clear.

  • We have revised the title of Figure 3 to a shorter formulation and clarified the content in the legend. The legend now specifies that the figure presents box-and-whisker plots of the relative error of the ARMIE PM5 measurements compared with the DustTrak and SMPS + APS reference instruments.

Discussion: Expand the discussion to include comparisons with similar work in the literature, especially studies using the SPS30 in residential or indoor settings.

  • Thank you for this suggestion. We have expanded the Discussion to place our findings in the context of previous evaluations of the SPS30. The revised text now references relevant work demonstrating similar performance characteristics when the SPS30 is compared with optical and mobility-based instruments [Line 425-428 & 433-435].

Reviewer 2 Report

Comments and Suggestions for Authors

The manuscript under consideration evaluates the performance of an optical low-cost PM sensor considering different reference instruments, indoor PM sources and calibration methods.

Although the aims and experimental setup of the study are certainly relevant and of interest to the research community, I'm missing an extensive discussion on the observed effects of humidity, source differences in residuals, reference instruments (detection principles) and the implications for real-world applications.

Major comments include:

  • The introduction should include more information and references on the current state-of-the-art of opticle low-cost PM sensors, past evaluation studies and their applications/added-value.
  • Why have authors not included road traffic or wood combustion-related PM in the experiments (most relevant in terms of personal exposure)? Please elaborate on this limitation within the manuscript
  • What is the targeted application of the sensor? Indoor/outdoor exposure? The temporal monitoring resolution of the ARMIE sensor is 5 minutes which is too low for mobile applications. Please elaborate on the envisaged use cases and associated strengths/weaknesses of the considered sensor system (considered pollutant sources/RH/concentration range/...).
  • Authors state that results show heteroscedasticity across comparison instruments, aerosol sources, as well as calibrated and uncalibrated data, after which they conclude that the sensor performs "with high correlation" when compared to the Dusttrak. The results section and Bland-Altmann plots show a significant trends and dependencies within the collected sensor data which is not properly targeted in the discussion section: e.g. the appied calibration models do not perform on the entire concentration range. How can you cope with this (non-linear models, multi-level calibration)? Different residual trends are observed when considering different sources; what are potential reasons for this (compositional,particle size range,...)?
  • How does the observed performance compare to other sensor evaluation studies. Authors state that a lot of sensor evaluation studies ar eavailable, please compare these findings to similar studies and differentiate the outcomes of this study to existing literature.
  • Different relative humidity conditions were considered as mentioned in the discussion, but not in the methods section. Please include in the methodology; procedure to test for RH, time needed to set RH conditions within the chamber, and evaluation method.
  • Only parametric calibration models (linear and multilinear) have been tested and showed to perform only partly. Did authors test for normality of the data distributions, test alternative calibration approaches,...?
  • Figure 1 reflects instrument differences (Dustrak vs APS/SMPS) in response time and PM concentrations. Please elaborate on these differences within the discussion section.
  • Ln 264: "Each point of Figure 2 is color-coded by sensor unit": In the Figure legend, every color represents a different source... Please clarify.
  • Figure 2: Can authors explain the observed residuals difference between of candles and cigarettes/cooking smoke? Please elaborate on potential compositional/particle distribution differences.
  • Figure 2: 

    Figures clearly indicate that the applied calibration strategy does not perform at higher >100 µg/m³ concentrations. Have the authors tested non-linear calibration models or alternative calibration approaches (multiple linear models depending on concentration range). Many calibration examples are described in literature, e.g.:

    https://doi.org/10.3390/atmos13060944 
    https://doi.org/10.5194/amt-13-1181-2020 
    https://doi.org/10.1080/02786826.2015.1100710 
    https://doi.org/10.3390/atmos10020041 
    https://doi.org/10.1021/acs.est.3c04843 
    https://www.mdpi.com/1660-4601/18/11/6007 

  • What is the relevance of Figure 3 and what does it mean in terms of sensor performance. Please discuss the implications of this Figure in the results and/or discussion section of remove this Figure
  • The conclusion section 

    is rather limited, especially after reporting the observed heteroscedasticity across comparison instruments and aerosol sources. 

    Please reflect on the original study aims put forward in the introduction:
    "- Characterize the ARMIE nodes' technical design and detection principles. 
    - Assess the accuracy of raw PM2.5 readings relative to a co‑located comparison and higher-grade instruments across different emission sources and environmental conditions, using controlled chamber experiments with aerosols from cooking, candles, and cigarette smoke. Evaluate calibration strategies to improve measurement agreement under varying conditions."

    Based on the experimental outcomes, what is the application potential for these sensors? I'm missing an extensive discussion with focus on the collected experimental evidence, the implications for the intended use case and comparison against existing sensor evaluation studies.

Minor comments:

  • Please mention the limitation of low-cost sensors technologies in detecting coarse sized particles:

    https://doi.org/10.3390/s24175653

    https://doi.org/10.5194/amt-13-2413-2020

    https://vaquums.eu/sensor-db/tests/life-vaquums_pmfieldtest.pdf/view

  • The size of the exposure chamber is rather large. Did authors evaluate the homogenity of PM concentration levels within the chamber and how were sensors and reference instruments positioned? What was the impact on response times?
  • Please include the mathematical functions to calculate the considered evaluation metrics.
  • Ln 360 Clear differences are observed between the applied reference instrumenst. Authors should elaborate more on different measurement principles of the considered instruments that might account for the observed discrepancies.
  • Ln 368: What do authors mean with this statement? Does sensor performance improve when you exclude the data from one sensor? This would imply large between sensor uncertainty... Please elaborate
  • Ln 375: why is this range relevant to epi research? Please substantiate with relevant REFs
  • Section 4.3: What are the authors conclusions related to RH sensitivity of the ARMIE sensor? This section should discuss the experimental RH results.
  • Ln 425: Is there a need for repeatable in generated PM2.5 peaks/concentrations? Rather, there is a need for repeatability between the individual sensor/REF responses and performance.
  • Please consider other minor comments and formatting comments within the attached PDF file.

Author Response

Major comments include:

  • The introduction should include more information and references on the current state-of-the-art of opticle low-cost PM sensors, past evaluation studies and their applications/added-value.
    •  We appreciate the emphasis on strengthening the contextual background for low-cost optical PM sensors. In line with this suggestion, and consistent with the revisions made following Reviewer 1’s feedback, we expanded an existing subsection in the Introduction to further clarify the application domains, added value, and performance characteristics reported in the literature. [Line 68-86].
  • Why have authors not included road traffic or wood combustion-related PM in the experiments (most relevant in terms of personal exposure)? Please elaborate on this limitation within the manuscript.
    • The sensor node is intended primarily for assessing indoor exposure, where individuals spend the majority of their time, and where personal activities substantially influence pollutant levels. For this reason, we focused on the most common indoor emission sources, cooking, candles, and cigarette smoke, which are both relevant for public-health applications and modifiable at the individual or household level. Traffic- and woodsmoke-related aerosols were therefore not included in the chamber experiments.
  • What is the targeted application of the sensor? Indoor/outdoor exposure? The temporal monitoring resolution of the ARMIE sensor is 5 minutes which is too low for mobile applications. Please elaborate on the envisaged use cases and associated strengths/weaknesses of the considered sensor system (considered pollutant sources/RH/concentration range/...).
    • We have added a clarification in the Materials and Methods [Line 116-118] describing the intended application of the ARMIE sensor-node. As noted in the revised text, the device is designed primarily for assessing indoor personal exposure patterns, where pollutant levels are strongly influenced by individual behaviours and microenvironments, and where a five-minute recording interval is generally adequate for capturing relevant exposure variability.
  • Authors state that results show heteroscedasticity across comparison instruments, aerosol sources, as well as calibrated and uncalibrated data, after which they conclude that the sensor performs "with high correlation" when compared to the Dusttrak. The results section and Bland-Altmann plots show a significant trends and dependencies within the collected sensor data which is not properly targeted in the discussion section: e.g. the appied calibration models do not perform on the entire concentration range. How can you cope with this (non-linear models, multi-level calibration)? Different residual trends are observed when considering different sources; what are potential reasons for this (compositional, particle size range,...)?
    • We have expanded the manuscript with a conclusion where we clarify that we found a high correlation with DustTrak, indicating a high precision of the sensors, but as seen in the Bland-Altman plots, a high heteroscedasticity [Line 596-618]. In particular, we now note that high correlation primarily reflects precision with the co-located reference instruments and does not assess accuracy relative to a reference instrument. To address this, we incorporated Bland–Altman analyses and examined concentration-dependent behavior, including the heteroscedasticity observed at increasing PM2.5 levels. The revised text further highlights the value of relative error and other scale-dependent metrics for evaluating sensor performance across the full concentration range.
  • How does the observed performance compare to other sensor evaluation studies. Authors state that a lot of sensor evaluation studies ar eavailable, please compare these findings to similar studies and differentiate the outcomes of this study to existing literature.
    • In revising the manuscript, we strengthened the Discussion to more clearly situate our findings within the existing evaluation literature, with particular emphasis on studies that have assessed the Sensirion SPS30 [Line 421-423 & 428-430]. We focused on SPS30-specific evaluations as these provide the most transferable points of comparison for our results, given that sensor performance is highly affected by the specific sensor model and manufacturer.
  • Different relative humidity conditions were considered as mentioned in the discussion, but not in the methods section. Please include in the methodology; procedure to test for RH, time needed to set RH conditions within the chamber, and evaluation method.
    • We have clarified the description of the environmental chamber in the Materials and Methods section. The revised text now specifies that the chamber is equipped with measurements for temperature, relative humidity, airflow and mixing-fan operation, which are displayed on a digital control panel [line 163-166]. We also describe the ceiling-mounted climate fan used to ensure adequate air mixing and to stabilize temperature and humidity during experiments.
  • Only parametric calibration models (linear and multilinear) have been tested and showed to perform only partly. Did authors test for normality of the data distributions, test alternative calibration approaches,...?
    • We acknowledge the limitation that we did not undertake a broader exploratory search for alternative or more complex calibration models. Such approaches may have improved the fit to the present dataset. However, we would also like to highlight an important constraint in relation to the intended use of the ARMIE node. Each participant and their home constitutes a distinct microenvironment with considerable variability in aerosol sources, ventilation, and background conditions. More flexible or highly optimized calibration models risk capturing environment-specific noise rather than true signal. In our view, this could increase the risk of exposure misclassification when the sensor-node is deployed in population-based monitoring campaigns. For this reason, we applied simple parametric calibration models that prioritize transparency and generalizability.
  • Figure 1 reflects instrument differences (Dustrak vs APS/SMPS) in response time and PM concentrations. Please elaborate on these differences within the discussion section.
    • We have added a clarification to the Discussion, noting that the discrepancies observed in Figure 1 between the DustTrak and the SMPS + APS vary by emission source [Line 546-550]. This reflects the fundamentally different measurement principles of optical photometry versus mobility- and aerodynamic-based particle sizing, as well as source-specific differences in aerosol composition and size distribution.
  • Ln 264: "Each point of Figure 2 is color-coded by sensor unit": In the Figure legend, every color represents a different source... Please clarify.
    • Thank you for pointing this out. The text has been corrected to specify that the points in Figure 2 are colour-coded by emission source, consistent with the figure legend.
  • Figure 2: Can authors explain the observed residuals difference between of candles and cigarettes/cooking smoke? Please elaborate on potential compositional/particle distribution differences.
    • We have expanded the Results section to clarify the source-dependent residual patterns observed in Figure 2. As described in the revised text [Line 319-325], candle emissions are dominated by soot aggregates with low optical scattering efficiency, leading to underestimation by the ARMIE sensor when compared with the SMPS + APS. In contrast, cooking and cigarette smoke generate predominantly organic particles produced at lower temperatures, which scatter light more effectively and therefore result in overestimation by the ARMIE sensor relative to the reference instruments.
  • Figure 2:  Figures clearly indicate that the applied calibration strategy does not perform at higher >100 µg/m³ concentrations. Have the authors tested non-linear calibration models or alternative calibration approaches (multiple linear models depending on concentration range). Many calibration examples are described in literature, e.g.:
    https://doi.org/10.3390/atmos13060944 
    https://doi.org/10.5194/amt-13-1181-2020 
    https://doi.org/10.1080/02786826.2015.1100710 
    https://doi.org/10.3390/atmos10020041 
    https://doi.org/10.1021/acs.est.3c04843 
    https://www.mdpi.com/1660-4601/18/11/6007 
    • Thank you for this comment, and for the suggested literature. We have expanded the Discussion [Line 489-495] to elaborate on hygroscopic growth and to summarise the evidence showing that RH-adjusted calibration can improve optical sensor performance under high-humidity conditions. With respect to alternative calibration approaches, we refer to our response to Comment 7, where we explain the rationale for limiting the calibration models in the present study and the considerations associated with more flexible or exploratory modelling strategies.

  • What is the relevance of Figure 3 and what does it mean in terms of sensor performance. Please discuss the implications of this Figure in the results and/or discussion section of remove this Figure
    • Figure 3, together with Appendix A Figure A2, highlights the concentration-dependent behavior of the ARMIE sensor. While error increases at higher concentrations, the relative error forms a more stable and interpretable pattern, particularly for comparisons with the DustTrak. This indicates that relative error may capture aspects of sensor performance that are obscured in error metrics and may therefore be useful in future calibration frameworks. We have not discussed this further, as we do not have any methodological suggestions to implement such relative errors into a calibration model.
  • The conclusion section is rather limited, especially after reporting the observed heteroscedasticity across comparison instruments and aerosol sources. 
    Please reflect on the original study aims put forward in the introduction:
    "- Characterize the ARMIE nodes' technical design and detection principles. 
    - Assess the accuracy of raw PM2.5 readings relative to a co‑located comparison and higher-grade instruments across different emission sources and environmental conditions, using controlled chamber experiments with aerosols from cooking, candles, and cigarette smoke. Evaluate calibration strategies to improve measurement agreement under varying conditions."
    Based on the experimental outcomes, what is the application potential for these sensors? I'm missing an extensive discussion with focus on the collected experimental evidence, the implications for the intended use case and comparison against existing sensor evaluation studies.
  • We have drafted a dedicated Conclusion section to the manuscript. The revised section summarizes how the findings address the original study objective 2 and 3 [Line 593-616]. We hope the Conclusion section sufficiently meets the intention of this comment.

Minor comments:

  • Please mention the limitation of low-cost sensors technologies in detecting coarse sized particles:
    • We have implemented a mention on this limitation [Line 107-110].
  • https://doi.org/10.3390/s24175653
  • https://doi.org/10.5194/amt-13-2413-2020
  • https://vaquums.eu/sensor-db/tests/life-vaquums_pmfieldtest.pdf/view
  • The size of the exposure chamber is rather large. Did authors evaluate the homogeneity of PM concentration levels within the chamber and how were sensors and reference instruments positioned? What was the impact on response times?
    • Thank you for this comment. All ARMIE units and reference instruments were colocated in the centre of the chamber during each experiment, and a ceiling-mounted climate fan was used continuously to promote air mixing and maintain stable conditions. We did not conduct a formal spatial homogeneity assessment, and we did not determine the Reynolds numbers for the airflow conditions, as also noted in the Methods section [Line 169-172]. However, the chamber is designed for controlled aerosol studies, and the use of continuous mixing together with colocated instrument placement minimises the likelihood of spatial gradients influencing the measurements.
  • Please include the mathematical functions to calculate the considered evaluation metrics.
    • The mathematical formulas used to calculate the evaluation metrics have now been added to Appendix C.
  • Ln 360 Clear differences are observed between the applied reference instrumenst. Authors should elaborate more on different measurement principles of the considered instruments that might account for the observed discrepancies.
    • Thank you for this comment. We have added a subsection to the discussion with a focus on the assumptions in SMPS + APS number to mass conversion [Line557-561] as well as the absence of gravimetric reference measurements [Line 562-567].
  • Ln 368: What do authors mean with this statement? Does sensor performance improve when you exclude the data from one sensor? This would imply large between sensor uncertainty... Please elaborate
    • The leave-one-sensor-out approach was used to assess the generalizability of the calibration models across different ARMIE units, not to identify or exclude underperforming sensors. The results do not indicate improved performance when a sensor is omitted; rather, this procedure functions as an internal validation method, evaluating how well calibration models perform on a sensor that was not included in the training data.
  • Ln 375: why is this range relevant to epi research? Please substantiate with relevant REFs.
    • As highlighted [Line 466-468], indoor PM5 concentrations generated by everyday activities such as cooking, candle burning and e-cigarette use frequently exceed the levels required to be 97.5% certain that actual concentrations surpass the WHO air-quality guidelines. This concentration range is therefore directly relevant for epidemiological research, as these indoor sources are major contributors to personal exposure and account for much of the within-person variability
  • Section 4.3: What are the authors conclusions related to RH sensitivity of the ARMIE sensor? This section should discuss the experimental RH results.
    • We have expanded the discussion section including a discussion for the intended use case of the ARMIE sensor in indoor personal exposure monitoring [Line 489-495] - where RH is typically moderate - the impact on performance is expected to be minimal. We hope that the expanded discussion regarding the intended use case of the ARMIE sensor meets the intention of the reviewer.

  • Ln 425: Is there a need for repeatable in generated PM2.5 peaks/concentrations? Rather, there is a need for repeatability between the individual sensor/REF responses and performance.
    • We agree that sensor performance should be evaluated in terms of the repeatability of the sensor responses relative to the reference instruments. Our discussion of reproducibility refers to the variability in the generated PM2.5 concentrations, which is difficult to eliminate even under controlled laboratory conditions and reflects the heterogeneity of real-world indoor emissions.

Reviewer 3 Report

Comments and Suggestions for Authors

Laboratory Evaluation of ARMIE, a Portable SPS30-Based Low-Cost Sensor Node for PM2.5 Monitoring

The authors present the results of the laboratory evaluation of an air quality monitor based on the SPS30 low-cost sensor for PM2.5 monitoring.

The low-cost sensor was collocated with an optical DustTrak photometer and a combination of a Scanning Mobility Particle Sizer (SMPS) and an Aerodynamic Particle Sizer (APS). The evaluation environment included exposures to candle burning, cooking, cigarette smoke and clean air, across normal and high-humidity conditions.

The authors contest that most of the studies for the evaluation of low-cost sensors for air quality monitoring have been concentrated in outdoor environments, unlike the present study.

The manuscript is well written [with the aid of ChatGPT, according to the authors], the methodology is thoroughly presented, the discussions are detailed, and figures, Tables and supplementary data are clear and informative.

The readers of Sensors should find the manuscript a useful addition to the field.

Suggestions for improvement:

The main drawback of the study is the lack of in-depth review of past similar studies and how they relate to the present one.

  • Introduction section [page 1] relies a good deal on Reference 9 [Raysoni, A.U., et al. 2023], a review paper. Whilst this is not inappropriate, the reader does not get the context of the work in terms of previous studies in this area. Thus:
    • Previous indoor evaluations of low-cost sensors should be reviewed in more detail.
    • Especially, it should be clear that there is a gap in their [previous studies] use of low-cost sensors in evaluating ‘realistic indoor emission sources’ as the authors suggest [lines 70 to 72].
  • This leads on to the Discussion section [page 14]; the authors should discuss the results of the work in relation to previous studies in greater detail. Thus:
    • How do the results defer [or not defer] from similar previous studies and what can be concluded from these outcomes?
    • In Section 4.8 [Perspectives] the authors suggest that low-cost sensors are unsuitable for absolute representation of PM2.5 values but can be used to evaluate how “everyday environments drive the variability in personal exposure.” They should elaborate on this, using references from literature. In particular, does this mean that “mid-cost instruments like the DustTrak” cannot be used for epidemiological studies especially where the actual values of the particulate matter concentration are required?

Author Response

Reviewer 3

Dear Reviewer,

Thank you very much for your careful evaluation of the manuscript. Before addressing your specific comments, we would like to offer one clarification regarding the calibration model used in the original submission. Following Reviewer 1’s remark about the missing calibration formula, we revisited the complete set of analysis functions and code. Through this review we identified that relative humidity (RH) had been implemented as an additive term rather than as an interaction with the ARMIE PM2.5 readings, which is the statistically appropriate specification given the underlying measurement behaviour.

We have therefore re-estimated all calibration models using an interaction term between ARMIE PM2.5 and RH. Although this represents an important methodological correction, its influence on the numerical results was limited. For the DustTrak-based calibration, the changes were confined to minor decimal-level differences. For the SMPS + APS calibration, the impact was more evident, but did not alter the interpretation of the results or the overall conclusions of the study.

All tables and figures have been updated accordingly, and the revised model specification is now clearly described in the manuscript.

We are grateful for the opportunity to correct this point and sincerely appreciate your contribution to improving the methodological clarity of the work.

Comments and Suggestions for Authors

Laboratory Evaluation of ARMIE, a Portable SPS30-Based Low-Cost Sensor Node for PM2.5 Monitoring

The authors present the results of the laboratory evaluation of an air quality monitor based on the SPS30 low-cost sensor for PM2.5 monitoring.

The low-cost sensor was collocated with an optical DustTrak photometer and a combination of a Scanning Mobility Particle Sizer (SMPS) and an Aerodynamic Particle Sizer (APS). The evaluation environment included exposures to candle burning, cooking, cigarette smoke and clean air, across normal and high-humidity conditions.

The authors contest that most of the studies for the evaluation of low-cost sensors for air quality monitoring have been concentrated in outdoor environments, unlike the present study.

The manuscript is well written [with the aid of ChatGPT, according to the authors], the methodology is thoroughly presented, the discussions are detailed, and figures, Tables and supplementary data are clear and informative.

The readers of Sensors should find the manuscript a useful addition to the field.

Suggestions for improvement:

The main drawback of the study is the lack of in-depth review of past similar studies and how they relate to the present one.

  • Introduction section [page 1] relies a good deal on Reference 9 [Raysoni, A.U., et al. 2023], a review paper. Whilst this is not inappropriate, the reader does not get the context of the work in terms of previous studies in this area. Thus:
    - Previous indoor evaluations of low-cost sensors should be reviewed in more detail.
    - Especially, it should be clear that there is a gap in their [previous studies] use of low-cost sensors in evaluating ‘realistic indoor emission sources’ as the authors suggest [lines 70 to 72].
    • We appreciate the emphasis on strengthening the contextual background for previous evaluations of low-cost optical PM sensors, including studies of the SPS30. In line with this suggestion, and consistent with the revisions made following Reviewer 2, we expanded the relevant subsection of the Introduction to clarify the application domains, added value and performance characteristics reported in the existing literature. These revisions [Line 65-81]of the manuscript, now provide a clearer basis for situating the present study within the broader field on the evaluation of the SPS30 sensor.
  • This leads on to the Discussion section [page 14]; the authors should discuss the results of the work in relation to previous studies in greater detail. Thus:
    - How do the results defer [or not defer] from similar previous studies and what can be concluded from these outcomes?
    • We have expanded the Discussion to situate our findings more clearly within the existing evidence base for the SPS30. The revised text includes comparisons with previous evaluations reporting similar performance characteristics when the SPS30 is assessed against optical instruments [Line 419-421], and weaker agreement when compared with mobility- or aerodynamic-based reference systems [Line 426-428].
  • In Section 4.8 [Perspectives] the authors suggest that low-cost sensors are unsuitable for absolute representation of PM2.5 values but can be used to evaluate how “everyday environments drive the variability in personal exposure.” They should elaborate on this, using references from literature. In particular, does this mean that “mid-cost instruments like the DustTrak” cannot be used for epidemiological studies especially where the actual values of the particulate matter concentration are required?
    • We have revised the subsection [lines 581–595] to clarify the distinction between precision and accuracy in the context of personal exposure monitoring. As noted in the revised text, the precision of low-cost sensors, even when accompanied by limited accuracy, can be sufficient for investigating whether fluctuations in personal PM5 exposure are associated with health outcomes. It is not our intention to imply that mid-cost instruments such as the DustTrak cannot be used; rather, their high cost and logistical demands make them unsuitable for large-scale population-based monitoring campaigns, where adequate statistical power requires monitoring many participants simultaneously.

Reviewer 4 Report

Comments and Suggestions for Authors

1. Recommendation
Major Revision
Title: Laboratory Evaluation of ARMIE, a Portable SPS30-Based Low-Cost Sensor Node for PM2.5 Monitoring
Overview and General Recommendations:
PM2.5 sensors range from high-end to low-end. However, low-end PM2.5 sensors often lack accuracy and environmental applicability compared to high-end sensors. Therefore, the development of a PM2.5 sensor that is both low-cost and highly accurate, as well as environmentally applicable, is essential. This paper evaluated ARMIE, a low-cost sensor, and compared it with existing high-precision equipment to identify its limitations and accuracy. Therefore, this paper can assist researchers in sensor selection and accuracy improvement research. However, several major revisions are needed.

This paper is well-written, with a relatively detailed introduction, methods, results, and discussion. Consequently, it is considered suitable for publication in Sensors after a major revision.

The main comments on the paper are as follows:
Abstract
The results are generally well-organized.

1. Introduction

Although a review of PM2.5 health effects, sensor performance evaluation studies, and low-cost sensor studies were conducted, they are insufficient to explain the performance evaluation of the ARMIE sensor in this paper. Additional reviews are recommended, including a performance comparison of optical PM2.5 sensors, a review of the effects of temperature and humidity, and a performance comparison of various sensor models. Therefore, additional reviews of related papers are recommended.

2. Materials and Methods

Lines 91, 105, and 112 require renumbering.
Line 170, "30L/S1" should be changed to "30L S-1."
Lines 152, 157, 165, 178, 192, and 203 require renumbering.

Overall, the test method is clearly and well-summarized.

3. Results

3.1. Data Overview
Explain which test results are obtained, and it is recommended to add Figure 1 at the end.

Also, the MDPI Figure format needs to be reviewed and revised.
The MDPI format for Table 1 needs to be reviewed.
Since the table is embedded within the paper, expressions such as Appendix A should be removed.

3.2 Leave-one-sensor-out
The MDPI Figure format needs to be reviewed and revised.
The MDPI format for Table needs to be reviewed.

3.3 Leave-one-sensor-out

The MDPI Figure format needs to be reviewed and revised.
The MDPI format for Table needs to be reviewed.
The titles for Sections 3.2 and 3.3 are the same. The titles need to be revised.

4. Discussion
The Discussion section is relatively well written.

The final Perspectives section should be the Conclusion section, and a conclusion summarizing the Results and Discussion should be included.

5. Conclusion

Needs to be written.

6. References

Needs to be rewritten in MDPI format.

The experimental content of the paper is relatively thorough and logical, but the format and conclusion require rewriting. Therefore, publication is possible after a major revision.

Author Response

Reviewer 4

Dear Reviewer,

Thank you very much for your careful evaluation of the manuscript. Before addressing your specific comments, we would like to offer one clarification regarding the calibration model used in the original submission. Following Reviewer 1’s remark about the missing calibration formula, we revisited the complete set of analysis functions and code. Through this review we identified that relative humidity (RH) had been implemented as an additive term rather than as an interaction with the ARMIE PM2.5 readings, which is the statistically appropriate specification given the underlying measurement behaviour.

We have therefore re-estimated all calibration models using an interaction term between ARMIE PM2.5 and RH. Although this represents an important methodological correction, its influence on the numerical results was limited. For the DustTrak-based calibration, the changes were confined to minor decimal-level differences. For the SMPS + APS calibration, the impact was more evident, but did not alter the interpretation of the results or the overall conclusions of the study.

All tables and figures have been updated accordingly, and the revised model specification is now clearly described in the manuscript.

We are grateful for the opportunity to correct this point and sincerely appreciate your contribution to improving the methodological clarity of the work.

Comments and Suggestions for Authors

  1. Recommendation
    Major Revision
    Title: Laboratory Evaluation of ARMIE, a Portable SPS30-Based Low-Cost Sensor Node for PM2.5 Monitoring
    Overview and General Recommendations:
    PM2.5 sensors range from high-end to low-end. However, low-end PM2.5 sensors often lack accuracy and environmental applicability compared to high-end sensors. Therefore, the development of a PM2.5 sensor that is both low-cost and highly accurate, as well as environmentally applicable, is essential. This paper evaluated ARMIE, a low-cost sensor, and compared it with existing high-precision equipment to identify its limitations and accuracy. Therefore, this paper can assist researchers in sensor selection and accuracy improvement research. However, several major revisions are needed.

This paper is well-written, with a relatively detailed introduction, methods, results, and discussion. Consequently, it is considered suitable for publication in Sensors after a major revision.

The main comments on the paper are as follows:
Abstract
The results are generally well-organized.

  1. Introduction

Although a review of PM2.5 health effects, sensor performance evaluation studies, and low-cost sensor studies were conducted, they are insufficient to explain the performance evaluation of the ARMIE sensor in this paper. Additional reviews are recommended, including a performance comparison of optical PM2.5 sensors, a review of the effects of temperature and humidity, and a performance comparison of various sensor models. Therefore, additional reviews of related papers are recommended.

  • Answer [We appreciate the emphasis on providing a clearer foundation for evaluating the performance of the ARMIE sensor. In line with this recommendation, and consistent with the revisions made following Reviewer 2’s comments, we have expanded the relevant subsection of the Introduction to more clearly describe the application domains, added value, and performance characteristics reported in previous evaluations of low-cost optical PM sensors, including studies focusing on the SPS30.]
  1. Materials and Methods

Lines 91, 105, and 112 require renumbering.
Line 170, "30L/S1" should be changed to "30L S-1."
Lines 152, 157, 165, 178, 192, and 203 require renumbering.

Overall, the test method is clearly and well-summarized.

  1. Results

3.1. Data Overview
Explain which test results are obtained, and it is recommended to add Figure 1 at the end.

Also, the MDPI Figure format needs to be reviewed and revised.
The MDPI format for Table 1 needs to be reviewed.
Since the table is embedded within the paper, expressions such as Appendix A should be removed.

3.2 Leave-one-sensor-out
The MDPI Figure format needs to be reviewed and revised.
The MDPI format for Table needs to be reviewed.

3.3 Leave-one-sensor-out

The MDPI Figure format needs to be reviewed and revised.
The MDPI format for Table needs to be reviewed.
The titles for Sections 3.2 and 3.3 are the same. The titles need to be revised.

  • Answer [Thank you for these detailed comments regarding formatting and structure. We have carefully revised the manuscript to address all formatting-related issues you identified. We hope that these revisions now fully align with the formatting standards outlined by the MDPI journal.]
  1. Discussion
    The Discussion section is relatively well written.

The final Perspectives section should be the Conclusion section, and a conclusion summarizing the Results and Discussion should be included.

  • Answer [We have added a dedicated Conclusion section to the manuscript (lines 596–619), in which the main findings from the Results and Discussion are summarised. We hope that this revision meets the intention of the comment and provides a clearer and more complete conclusion to the study.]
  1. Conclusion

Needs to be written.

  1. References

Needs to be rewritten in MDPI format.

  • Answer [We have updated the references section to meet the formatting standards by the MDPI journal.]

The experimental content of the paper is relatively thorough and logical, but the format and conclusion require rewriting. Therefore, publication is possible after a major revision.

Round 2

Reviewer 1 Report

Comments and Suggestions for Authors

The authors have addressed my comments!

Author Response

Dear Reviewer,

Thank you for taking the time to review our manuscript and for providing constructive comments and suggestions. We appreciate your careful evaluation and agree that your feedback has helped to improve the clarity and presentation of our work.

In this revision round, we have specifically addressed the comments related to the presentation and structure of the Figures and Tables. We acknowledge that these elements had limitations in the previous version and required revision. All figures and tables have now been updated accordingly to improve clarity, consistency, and interpretability.

We hope that these revisions meet the standards expected by the reviewers and the journal.

On behalf of all authors, we thank you for your time and valuable feedback.

Reviewer 2 Report

Comments and Suggestions for Authors

Dear authors,

Thanks for considering my remarks and considerations. I believe the manuscript has improved and gained clarity.  However, I feel that some rebuttals/answers need further clarification.

See below:

Original comment: Authors state that results show heteroscedasticity across comparison instruments, aerosol sources, as well as calibrated and uncalibrated data, after which they conclude that the sensor performs "with high correlation" when compared to the Dusttrak. The results section and Bland-Altmann plots show a significant trends and dependencies within the collected sensor data which is not properly targeted in the discussion section: e.g. the appied calibration models do not perform on the entire concentration range. How can you cope with this (non-linear models, multi-level calibration)? Different residual trends are observed when considering different sources; what are potential reasons for this (compositional, particle size range,...)?

Answer authors: We have expanded the manuscript with a conclusion where we clarify that we found a high correlation with DustTrak, indicating a high precision of the sensors, but as seen in the Bland-Altman plots, a high heteroscedasticity [Line 596-618]. In particular, we now note that high correlation primarily reflects precision with the co-located reference instruments and does not assess accuracy relative to a reference instrument. To address this, we incorporated Bland–Altman analyses and examined concentration-dependent behavior, including the heteroscedasticity observed at increasing PM2.5 levels. The revised text further highlights the value of relative error and other scale-dependent metrics for evaluating sensor performance across the full concentration range.

Correlation does not reflect precision (between-sensor-uncertainty), but association with the REF instrument (while not necessarily being accurate). Please correct or clarify.

Original comment: Only parametric calibration models (linear and multilinear) have been tested and showed to perform only partly. Did authors test for normality of the data distributions, test alternative calibration approaches,...?

Answer authors: We acknowledge the limitation that we did not undertake a broader exploratory search for alternative or more complex calibration models. Such approaches may have improved the fit to the present dataset. However, we would also like to highlight an important constraint in relation to the intended use of the ARMIE node. Each participant and their home constitutes a distinct microenvironment with considerable variability in aerosol sources, ventilation, and background conditions. More flexible or highly optimized calibration models risk capturing environment-specific noise rather than true signal. In our view, this could increase the risk of exposure misclassification when the sensor-node is deployed in population-based monitoring campaigns. For this reason, we applied simple parametric calibration models that prioritize transparency and generalizability.

Please include this as calibration assumption within the manuscript. The impact from different microenvironments (RH and source impact) was experimentally evaluated within this study. As such the experimental results can substantiate the assumption of real-world uncertainty of low-cost sensors. 

Author Response

Dear Reviewer,

Thank you for taking the time to review our manuscript and for providing constructive comments and suggestions. We appreciate your careful evaluation and agree that your feedback has helped to improve the clarity and presentation of our work.

In this revision round, we have specifically addressed the comments related to the presentation and structure of the Figures and Tables. We acknowledge that these elements had limitations in the previous version and required revision. All figures and tables have now been updated accordingly to improve clarity, consistency, and interpretability.

We hope that these revisions meet the standards expected by the reviewers and the journal.

On behalf of all authors, we thank you for your time and valuable feedback.

Comment: [Original comment: Authors state that results show heteroscedasticity across comparison instruments, aerosol sources, as well as calibrated and uncalibrated data, after which they conclude that the sensor performs "with high correlation" when compared to the Dusttrak. The results section and Bland-Altmann plots show a significant trends and dependencies within the collected sensor data which is not properly targeted in the discussion section: e.g. the appied calibration models do not perform on the entire concentration range. How can you cope with this (non-linear models, multi-level calibration)? Different residual trends are observed when considering different sources; what are potential reasons for this (compositional, particle size range,...)?

Answer authors: We have expanded the manuscript with a conclusion where we clarify that we found a high correlation with DustTrak, indicating a high precision of the sensors, but as seen in the Bland-Altman plots, a high heteroscedasticity [Line 596-618]. In particular, we now note that high correlation primarily reflects precision with the co-located reference instruments and does not assess accuracy relative to a reference instrument. To address this, we incorporated Bland–Altman analyses and examined concentration-dependent behavior, including the heteroscedasticity observed at increasing PM2.5 levels. The revised text further highlights the value of relative error and other scale-dependent metrics for evaluating sensor performance across the full concentration range.

Correlation does not reflect precision (between-sensor-uncertainty), but association with the REF instrument (while not necessarily being accurate). Please correct or clarify. ]

  • Answer: [Thank you for this important clarification. We agree that a high correlation with a reference instrument should not be interpreted as evidence of sensor precision, nor as evidence of measurement accuracy. Rather, correlation reflects the strength of association between paired measurements and does not capture bias, heteroscedasticity, or concentration-dependent error.
    We have therefore revised the Conclusion section [Lines 600–629] to explicitly distinguish correlation from accuracy and to emphasise that, despite strong correlation with the DustTrak, substantial heteroscedasticity and concentration-dependent disagreement remain, as demonstrated by the Bland–Altman analyses. We believe this clarification addresses the reviewer’s concern and avoids drawing unsupported performance conclusions.]

Comment: [Original comment: Only parametric calibration models (linear and multilinear) have been tested and showed to perform only partly. Did authors test for normality of the data distributions, test alternative calibration approaches,...?

Answer authors: We acknowledge the limitation that we did not undertake a broader exploratory search for alternative or more complex calibration models. Such approaches may have improved the fit to the present dataset. However, we would also like to highlight an important constraint in relation to the intended use of the ARMIE node. Each participant and their home constitutes a distinct microenvironment with considerable variability in aerosol sources, ventilation, and background conditions. More flexible or highly optimized calibration models risk capturing environment-specific noise rather than true signal. In our view, this could increase the risk of exposure misclassification when the sensor-node is deployed in population-based monitoring campaigns. For this reason, we applied simple parametric calibration models that prioritize transparency and generalizability.

Please include this as calibration assumption within the manuscript. The impact from different microenvironments (RH and source impact) was experimentally evaluated within this study. As such the experimental results can substantiate the assumption of real-world uncertainty of low-cost sensors. 

  • Answer: [Thank you for the comment. We have now explicitly stated the underlying calibration assumption in the Methods section (Lines 247–253), clarifying the rationale for applying parsimonious parametric calibration models in the context of heterogeneous personal exposure settings. We hope that this addition meets the reviewer’s intention.]

Reviewer 3 Report

Comments and Suggestions for Authors

I thank the authors for responding to the comments. The manuscript is much better now, and I recommend it for acceptance by the editors.

Author Response

(The authors gave the same response as above.)

Reviewer 4 Report

Comments and Suggestions for Authors

The author has modified most of the previously requested changes. As long as the figure and reference formats are modified, publishing to Sensor should be possible without any issues.

Author Response

(The authors gave the same response as above.)
